# A subset of ipRGCs regulates both maturation of the circadian clock and segregation of retinogeniculate projections in mice

Kylie S Chew[1,2†], Jordan M Renna[3†], David S McNeill[1], Diego C Fernandez[1‡], William T Keenan[1], Michael B Thomsen[1], Jennifer L Ecker[1], Gideon S Loevinsohn[4], Cassandra VanDunk[5,6], Daniel C Vicarel[3], Adele Tufford[7], Shijun Weng[4], Paul A Gray[5,8], Michel Cayouette[7,9], Erik D Herzog[10], Haiqing Zhao[1], David M Berson[4*], Samer Hattar[1*‡]

[1]Department of Biology, Johns Hopkins University, Baltimore, United States; [2]Department of Biology, Stanford University, Stanford, United States; [3]Department of Biology, Program in Integrated Bioscience, The University of Akron, Akron, United States; [4]Department of Neuroscience, Brown University, Providence, United States; [5]Department of Anatomy and Neurobiology, Washington University, St. Louis, United States; [6]Department of Neurobiology, Northwestern University, Evanston, United States; [7]Cellular Neurobiology Research Unit, Institut De Recherches Cliniques De Montréal, Montreal, Canada; [8]Indigo Agriculture, Charlestown, United States; [9]Faculty of Medicine, Université De Montréal, Montreal, Canada; [10]Department of Biology, Washington University, St. Louis, United States

*For correspondence:
david_berson@brown.edu (DMB);
shattar@jhu.edu (SH)

†These authors contributed equally to this work

Present address: ‡Section on Light and Circadian Rhythms, National Institute of Mental Health, National Institute of Health, Bethesda, United States

Competing interests: The authors declare that no competing interests exist.

**Abstract** The visual system consists of two major subsystems, image-forming circuits that drive conscious vision and non-image-forming circuits for behaviors such as circadian photoentrainment. While historically considered non-overlapping, recent evidence has uncovered crosstalk between these subsystems. Here, we investigated shared developmental mechanisms. We revealed an unprecedented role for light in the maturation of the circadian clock and discovered that intrinsically photosensitive retinal ganglion cells (ipRGCs) are critical for this refinement process. In addition, ipRGCs regulate retinal waves independent of light, and developmental ablation of a subset of ipRGCs disrupts eye-specific segregation of retinogeniculate projections. Specifically, a subset of ipRGCs, comprising ~200 cells and which project intraretinally and to circadian centers in the brain, are sufficient to mediate both of these developmental processes. Thus, this subset of ipRGCs constitute a shared node in the neural networks that mediate light-dependent maturation of the circadian clock and light-independent refinement of retinogeniculate projections.

## Introduction

Light is detected by the eye for image-forming functions, including conscious perception of the visual scene, and for non-image-forming (NIF) functions, such as synchronization of circadian rhythms to the solar day (circadian photoentrainment) and the pupillary light reflex. These diverse visual functions, both image- and non-image-forming, require the retina for the detection and the initial processing of light signals, which are then relayed to the brain via the output neurons of the eye, retinal

ganglion cells (RGCs). The majority of RGCs project to image-forming centers in the brain, such as the dorsal lateral geniculate nucleus (dLGN) and the superior colliculus (SC). A subset of RGCs, however, is intrinsically photosensitive (*Berson, 2003*; *Berson et al., 2002*; *Hattar et al., 2002*; *Provencio et al., 1998*), in addition to receiving indirect light signals from the classical photoreceptors, rods and cones (*Lucas et al., 2003*; *Mrosovsky and Hattar, 2003*; *Panda et al., 2002*; *Ruby et al., 2002*; *Schmidt et al., 2008*). These intrinsically photosensitive RGCs (ipRGCs) constitute the sole conduit of light information to non-image-forming centers in the brain, such as the suprachiasmatic nucleus (SCN) (*Güler et al., 2008*; *Hattar et al., 2003*; *Panda et al., 2003*). ipRGCs drive non-image-forming behaviors even in the absence of rods and cones (*Czeisler et al., 1995*; *Foster et al., 1991*; *Freedman et al., 1999*; *Lucas et al., 2001, 1999*). Early reports envisioned a strict separation between the image- and non-image-forming visual networks (*Dreher and Robinson, 1991*; *Moore, 1997*; *Young and Lund, 1994*), but this view has begun to be challenged by recent evidence for functional crosstalk between the two systems (*Ecker et al., 2010*; *Estevez et al., 2012*; *Renna et al., 2011*; *Schmidt et al., 2014*; *Zhang et al., 2008*).

During development, all RGCs must achieve precise central connections in the brain that are necessary for the generation of visual behaviors. Precise visual circuits emerge in a multistep process: axon guidance pathways establish a coarse level of organization, which are then refined in an activity-dependent manner (*Arroyo and Feller, 2016*; *Oster and Sretavan, 2003*; *Osterhout et al., 2014*; *Wong, 1999*). The image-forming visual system has been a classic model of activity-dependent refinement of neuronal circuits, and both light-dependent and independent mechanisms play critical roles (*Hubel et al., 1977*; *Meister et al., 1991*; *Shatz and Stryker, 1988*). Refinement of the coarse projections of the RGCs to the dLGN is dependent on spontaneously generated neural activity, termed retinal waves. These retinal waves, with defined properties, sweep across the retina to instruct segregation of eye-specific projections to the dLGN and SC (*Ackman et al., 2012*; *Chandrasekaran et al., 2005*; *Feller, 2002, 2009*; *Firth et al., 2005*; *McLaughlin et al., 2003*; *Meister et al., 1991*; *Mrsic-Flogel et al., 2005*; *Muir-Robinson et al., 2002*; *Shatz and Stryker, 1988*; *Stellwagen and Shatz, 2002*; *Wong, 1999*; *Xu et al. 2011*; *Zhang et al., 2011*). Interestingly, light detection through ipRGCs influences retinal waves (*Renna et al., 2011*), although, eye-specific segregation still proceeds normally in the absence of light (*Demas et al., 2006*). In turn, retinal waves have been shown to modulate the intraretinal gap junction network of ipRGCs (*Arroyo et al., 2016*). Furthermore, light detection by ipRGCs has also been implicated in regulating developmental vascularization in the eye (*Rao et al., 2013*). However, it has not been determined whether ipRGCs have light-independent developmental roles or whether the developmental roles of ipRGCs have permanent functional consequences.

ipRGCs are best known for their ability to synchronize the circadian clock in the SCN to the solar day (*Chen et al., 2011*; *Güler et al., 2008*), a process known as circadian photoentrainment. The circadian clock contains an intrinsic genetic program that, in the absence of environmental light input, produces molecular and physiological rhythms with periods close to, but not exactly, 24 hr (*Harmer et al., 2001*; *Menaker et al., 1978*). Due to the autonomous nature of the circadian clock, the prevailing view has been that the fundamental features of the clock in the SCN, such as period length, do not require environmental input for maturation (*Davis and Menaker, 1981*; *Davis and Gorski, 1985*; *Jud and Albrecht, 2006*; *Pittendrigh, 1954*; *Richter, 1971*; *Vallone et al., 2007*; *Yamazaki et al., 2002*). However, there is intriguing evidence that animals that do not form eyes or the optic nerve due to genetic defects exhibit a lengthened circadian period (*Laemle and Ottenweller, 1998*; *Wee et al., 2002*). Since ipRGCs are the major, if not the sole, source of retinal input to the SCN (*Baver et al., 2008*; *Berson, 2003*; *Hattar et al., 2006, 2002*), these observations implicate a possible role for ipRGCs in the maturation of the circadian clock.

ipRGCs have now been shown to also project to the image-forming visual system (*Ecker et al., 2010*). These projections appear to arise from subtypes of ipRGCs that are morphologically and physiologically distinct from the originally identified ipRGCs, which are now known as M1 ipRGCs (*Ecker et al., 2010*; *Schmidt et al., 2011*; *Schmidt and Kofuji, 2009*; *Schmidt et al., 2008*). M1 ipRGCs predominantly innervate non-image-forming centers and are molecularly differentiated based on the expression of a transcription factor, *Pou4f2* (also called Brn3b) (*Chen et al., 2011*). Specifically, *Pou4f2*-negative M1 ipRGCs project exclusively to circadian centers in the brain and send intraretinal axonal collaterals (*Chen et al., 2011*), whereas the majority of M1 ipRGCs and all non-M1 ipRGCs express *Pou4f2*. The non-M1 ipRGC subtypes (M2-M5) send their axons to the

dLGN and SC and are capable of supporting coarse pattern vision in animals lacking functional rod and cone phototransduction pathways (*Brown et al., 2010*; *Ecker et al., 2010*). In addition, the M4 subtype of ipRGCs has been recently shown to influence vision by specifically mediating contrast sensitivity (*Estevez et al., 2012*; *Schmidt et al., 2014*).

Here, we used a variety of mutant mouse lines in which different proportions of ipRGCs are ablated at different ages to investigate developmental roles for ipRGCs and the long-term functional consequences. We show that a subset of ipRGCs (*Pou4f2*-negative M1 ipRGCs) influence the development of both the circadian and the visual systems. Furthermore, we show that light plays an essential role in the circadian but not the image-forming visual process.

## Results

### Developmental ablation of ipRGCs using Diphtheria toxin A subunit

We utilized two allelic changes at the melanopsin locus to ablate ipRGCs at different ages. As we have published previously, expression of attenuated diphtheria toxin (aDTA) from the melanopsin (*Opn4*) locus (*Opn4$^{aDTA}$*) results in ablation of mostly the M1 ipRGCs and does so only at adult ages (*Güler et al., 2008*) (*Figure 1A,B,D,E*). A newly generated line, expresses the full-strength version of diphtheria toxin (DTA) (*Opn4$^{DTA}$*) and causes ablation of ipRGCs at early postnatal stages (*Figure 1*; *Figure 1—figure supplement 1A–B*; *Supplementary file 1*). To quantify ipRGC loss in the aDTA and DTA lines, we used two genetic labeling methods. The *Opn4$^{Cre}$; Z/AP* mice, in which alkaline phosphatase (AP) expression is dependent on Cre expression in ipRGCs, labels all subtypes of ipRGCs (*Ecker et al., 2010*). Whereas in *Opn4$^{LacZ}$* mice only M1 ipRGCs are labeled following X-Gal staining for *β*-galactosidase activity (*Hattar et al., 2002*; *McNeill et al., 2011*). In animals heterozygous for DTA and Cre (*Opn4$^{Cre/DTA}$; Z/AP* mice; *Supplementary file 1*), ipRGCs were reduced in number at birth (*Figure 1A,C*). Total ipRGC number declined until P14, when approximately 500 cells survived, and then remained constant thereafter through 1year of age (*Figure 1A,C and E*). Using the LacZ locus with the aDTA or DTA loci, we show that at 6 months of age, about 75 M1 ipRGCs survived in *Opn4$^{LacZ/DTA}$* mice (*Supplementary file 1*), and consistent with our previous report, about 125 M1 ipRGCs survived in *Opn4$^{LacZ/aDTA}$* mice (*Güler et al., 2008*) (*Figure 1F*; *Supplementary file 1*). These results show that even some ipRGCs that express high levels of melanopsin (M1s) can survive the presence of a single dose of the full strength DTA.

We then sought to determine whether two alleles of DTA would result in a more complete ipRGC ablation. Since melanopsin is the only known marker for all ipRGCs, either an allele of Cre or LacZ is required at the melanopsin locus in order for ipRGCs to be labeled and quantified. Neither an anti-melanopsin antibody nor *in situ* hybridization with a melanopsin probe can be utilized because Cre, LacZ, aDTA, and DTA replace the melanopsin gene and thus result in a knockout for melanopsin. However, retinal innervation of the SCN originates exclusively from ipRGCs and thus the extent of innervation can be used as a proxy for ipRGC loss. To determine whether two copies of DTA (*Opn4$^{DTA/DTA}$*; *Supplementary file 1*) would produce further ablation of ipRGCs, we therefore examined retinal innervation of the SCN as an indirect measure of ipRGC loss. We labeled all retinal projections by injecting fluorescently labeled cholera toxin subunit*β* (CTB, a neuronal tracer) into the eyes and compared SCN innervation in 6-month-old wild-type, heterozygous (*Opn4$^{DTA/+}$*), homozygous attenuated-DTA (*Opn4$^{aDTA/aDTA}$*), and homozygous DTA (*Opn4$^{DTA/DTA}$*) mice (*Figure 1D*; *Supplementary file 1*). Sparse retinal fibers were found in the SCN of heterozygous DTA mice (*Opn4$^{DTA/+}$*) as observed in *Opn4$^{Cre/DTA}$; Z/AP* mice (*Figure 1A*) but these retinal fibers were entirely absent in *Opn4$^{DTA/DTA}$* mice (*Figure 1D*) suggesting that *Opn4$^{DTA/DTA}$* mice have more extensive ipRGC loss than *Opn4$^{DTA/+}$* mice. The more substantial and earlier ipRGC loss in *Opn4$^{DTA/DTA}$* mice was also confirmed by examining SCN innervation using CTB injections in P7 wild-type, *Opn4$^{aDTA/aDTA}$*, and *Opn4$^{DTA/DTA}$* mice (*Figure 1B*).

Given the strength of DTA as a toxin, we assessed *Opn4$^{DTA/DTA}$* mice for off-target effects. General retinal structure was evaluated by staining retinal sections from wild-type and *Opn4$^{DTA/DTA}$* mice with hematoxylin and eosin stains and additionally fluorescent staining for various retinal cell types, including cones, ON bipolar cells, calretinin-positive amacrine and ganglion cells, and Brn3a positive ganglion cells (*Figure 1—figure supplement 1C,E–F*). Wild-type and *Opn4$^{DTA/DTA}$* mice appeared similar by these stains, and quantification of total retinal thickness, as well as the thickness of each of

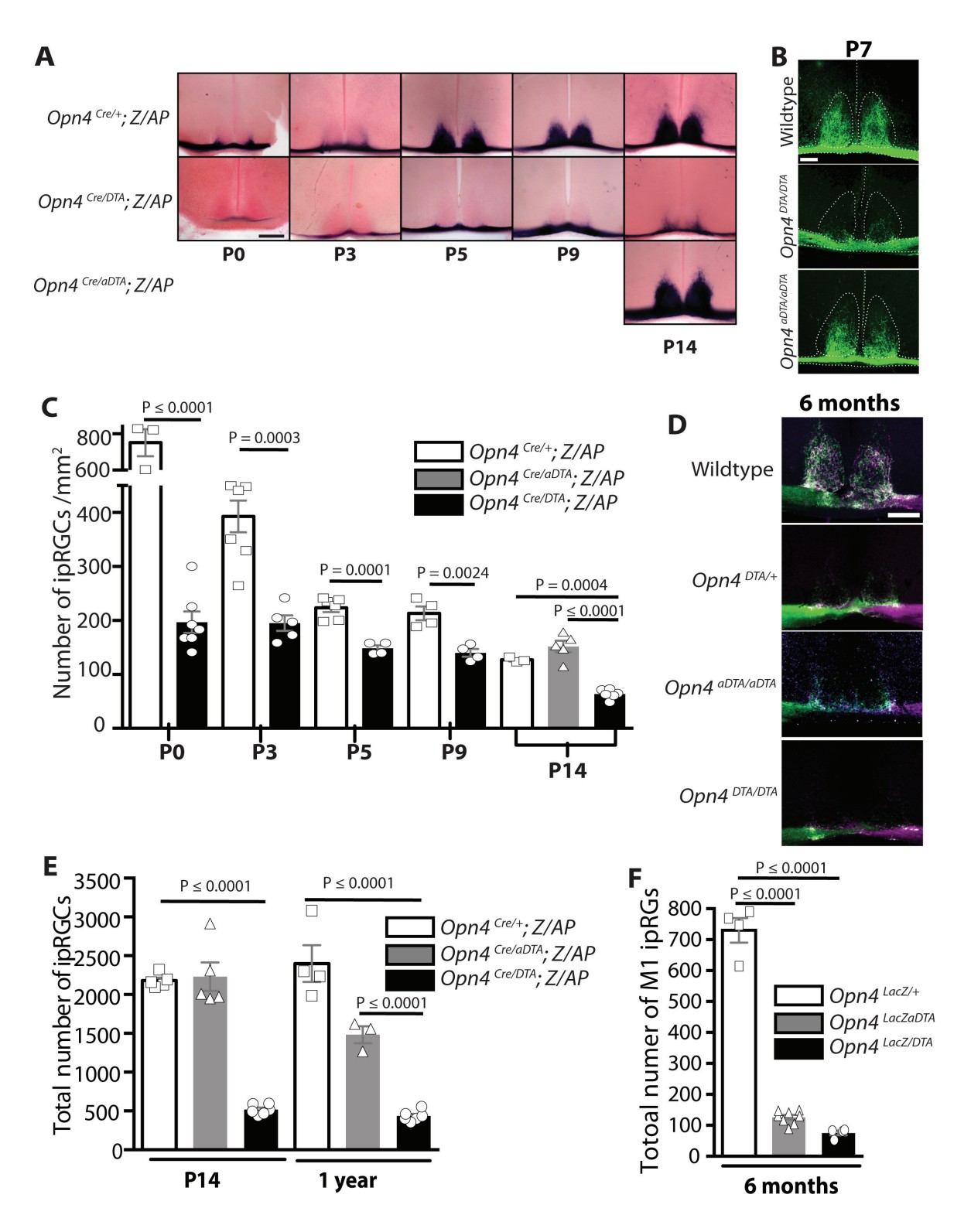

**Figure 1.** Developmental ablation of ipRGCs in the mouse retina. (**A**) Developmental time course of ipRGC innervation of the SCN, visualized by AP staining, in *Opn4^{Cre/+}; Z/AP* and *Opn4^{Cre/DTA}; Z/AP* mouse. For comparison, SCN staining from *Opn4^{Cre/aDTA}; Z/AP* mice at P14 are also shown. Scale bar = 200 μm. (**B**) SCN innervation in P7 WT, *Opn4^{DTA/DTA}*, and *Opn4^{aDTA/aDTA}* mice revealed by CTB injections into the eyes. Scale bar = 100 μm. (**C**) Developmental time course of ipRGC (all subtypes) cell density visualized by AP staining of retina from *Opn4^{Cre/+} Z/AP* (control) and *Opn4^{Cre/DTA}; Z/AP*

*Figure 1 continued on next page*

Figure 1 continued

mice at P0 (control n = 3, DTA n = 7), P3 (control n = 7, DTA n = 5), P5 (control n = 6, DTA n = 4), P9 (control n = 4, DTA n = 4), and P14 (control n = 3, DTA n = 6, aDTA n = 5). Cell counts from P14 retinas of $Opn4^{Cre/aDTA}$; Z/AP mice are also shown for comparison. Using a two-way ANOVA, we found a strongly significant effect of genotype. A t-test for P0, P3, P5, and P9 time points, and a one-way ANOVA with Bonferroni's post-hoc analysis for P14 revealed a significant cell loss at each time point. (D) SCN innervation revealed by CTB injections into the eyes of 6-month-old WT, $Opn4^{DTA/+}$, $Opn4^{DTA/DTA}$, and $Opn4^{aDTA/aDTA}$ mice. Scale bar = 200 μm. (E) Total cell counts of ipRGCs (all subtypes) revealed by alkaline phosphatase staining at P14 and 1 year of age in $Opn4^{Cre/+}$; Z/AP (control; P14: n = 5, 1 year: n = 4), $Opn4^{Cre/aDTA}$; Z/AP (P14: n = 5; 1 year: n = 3), and $Opn4^{Cre/DTA}$; Z/AP (P14: n = 6; 1 year: n = 6). Two-way ANOVA, Bonferroni's multiple comparisons test and adjusted p values. (F) Total cell counts of M1 ipRGCs, identified by x-Gal staining of retinas from 6 month old $Opn4^{LacZ/+}$ (control; n = 4), $Opn4^{LacZ/aDTA}$, and $Opn4^{LacZ/DTA}$ mice (n = 4). One-way ANOVA, Bonferroni's multiple comparisons test and adjusted p values. Error bars represent s.e.m. for all graphs. See also *Figure 1—figure supplement 1*.

The following figure supplements are available for figure 1:

**Figure supplement 1.** Generation and characterization of mice with an $Opn4^{DTA}$ allele.

**Figure supplement 2.** The number of cells in the SCN is unaffected in $Opn4^{DTA/DTA}$ mice.

the retinal layers, based on hematoxylin and eosin staining showed no difference between wild-type and $Opn4^{DTA/DTA}$ mice (*Figure 1—figure supplement 1D*). We also counted the number of SMI-32-positive ganglion cells, Brn3a-positive ganglion cells, cones, and starburst amacrine cells (*Figure 1—figure supplement 1G–J*). When we compared wild-type and $Opn4^{DTA/DTA}$ mice, there was no difference in cell number of any of these cell types. It is relevant to mention that about 50% of SMI-32 positive ganglion cells are actually M4 ipRGCs (*Estevez et al., 2012*; *Schmidt et al., 2014*), which express lower levels of melanopsin compared to M1 ipRGCs (*Ecker et al., 2010*; *Sexton et al., 2015*). Thus, expression of two copies of DTA from the melanopsin locus does not even succeed in killing all ipRGCs. Nonetheless, since we lack specific molecular markers for the other non-M1 ipRGC subtypes, we have no means of knowing the exact time course and extent of ablation for each of those subtypes. To assess potential off target effects in relevant brain regions, we counted the total number of nuclei (based on DAPI staining) and the total number of neurons (identified by antibody staining for Hu) in the SCN of wild-type and $Opn4^{DTA/DTA}$ mice. There was no difference between wild-type and $Opn4^{DTA/DTA}$ mice in the number of nuclei or neurons when compared in total or section by section through the SCN (*Figure 1—figure supplement 2*). Furthermore, the total size of the dLGN was also not different between wild-type and $Opn4^{DTA/DTA}$ mice or among any other tested mouse lines (*Figure 4—figure supplement 4E*). These data combined demonstrate that DTA efficiently and selectively ablated ipRGCs at early postnatal ages.

## ipRGCs are necessary to set the period length of the circadian clock

We recorded wheel-running activity under a 12:12-LD cycle, to measure circadian photoentrainment, and under constant darkness, to measure the intrinsic properties of the circadian clock (*Figure 2A–C*). We assessed wild-type and $Opn4^{LacZ/LacZ}$ (a null allele for melanopsin) mice as controls and also recorded wheel-running activity of $Opn4^{aDTA/aDTA}$, $Opn4^{DTA/DTA}$, $Opn4^{DTA/+}$, and $Opn4^{DTA/LacZ}$ mice (*Figure 2A*; *Figure 2—figure supplement 1*; *Supplementary file 1*). $Opn4^{LacZ/LacZ}$ and wild-type mice showed normal photoentrainment, phase shifting (at CT16), and free-running periods (*Figure 2A–C*). As previously published, $Opn4^{aDTA/aDTA}$ mice were unable to photoentrain or phase shift and free-ran under all lighting conditions with a period comparable to controls (*Güler et al., 2008*) (*Figure 2A–C*). $Opn4^{DTA/+}$ mice photoentrained, phase shifted (at CT16), and free-ran in constant darkness with a normal circadian period (*Figure 2A–C*) indicating that residual retinal input to the SCN in heterozygotes is sufficient for circadian photoentrainment. In the $Opn4^{DTA/LacZ}$ mice, the ~500 remaining ipRGCs (~75 M1 ipRGCs) in heterozygous DTA animals, further lose their intrinsic photoresponses due to the loss of both melanopsin alleles and thus can only relay rod/cone input to the brain. In these mice, circadian responses to light were highly attenuated, with inconsistent entrainment and phase-shifting, and exhibited a free-running period comparable to control mice (*Figure 2A–C*, *Figure 2—figure supplement 1A*). Similar to $Opn4^{aDTA/aDTA}$ mice, $Opn4^{DTA/DTA}$ mice did not photoentrain; however, they free-ran with an abnormally lengthened period (*Figure 2A,B*; *Figure 2—figure supplement 1B*). Because full-strength DTA, but not aDTA, kills ipRGCs during

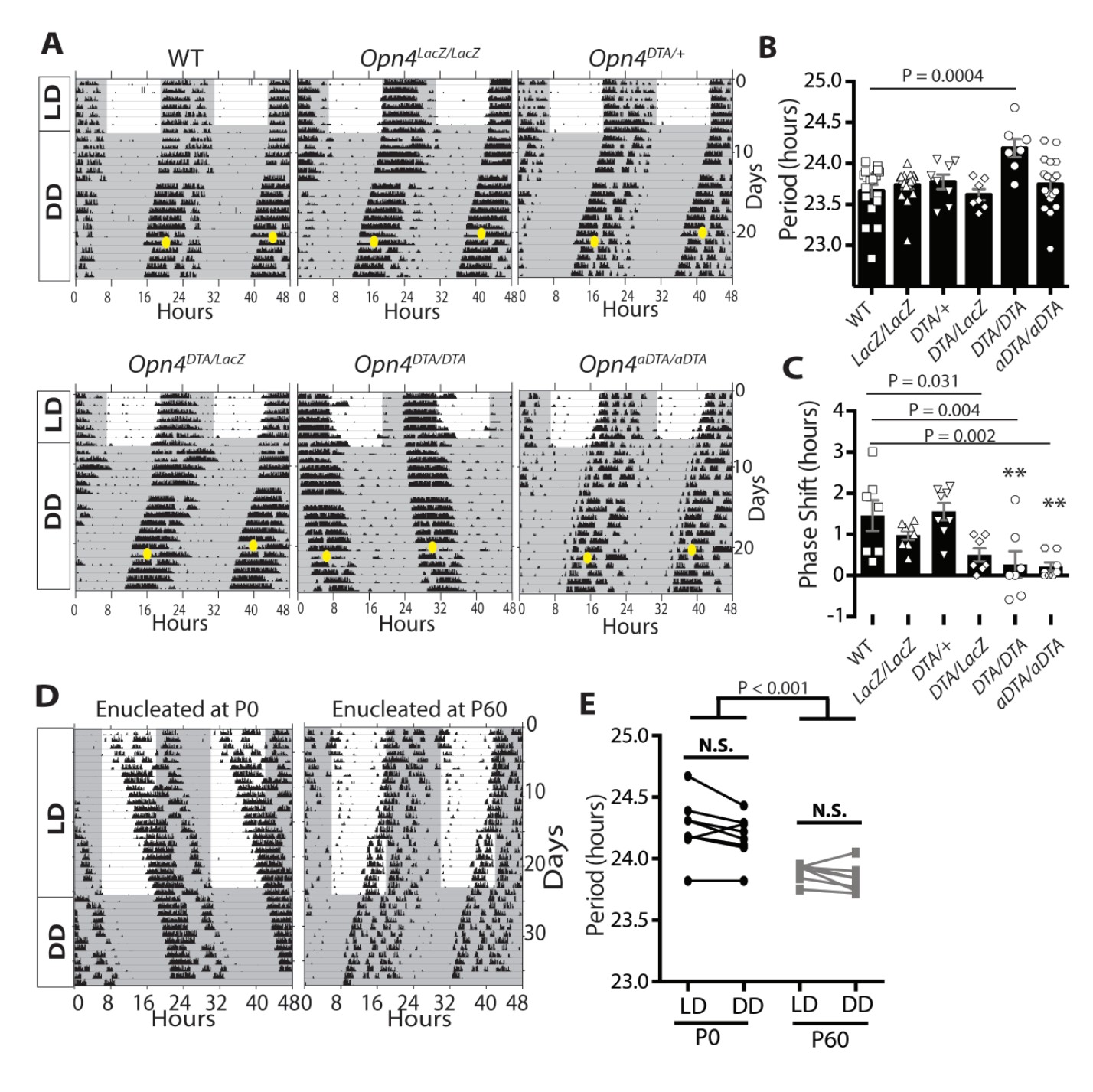

**Figure 2.** Developmental ablation of ipRGCs results in a lengthened circadian period. (**A**) Representative actograms of wild-type, *Opn4^LacZ/LacZ^*, *Opn4^DTA/+^*, *Opn4^DTA/LacZ^*, *Opn4^DTA/DTA^* and *Opn4^aDTA/aDTA^* mice under 12:12-LD cycle and constant darkness (DD). White background indicates light and grey background indicates darkness. (**B**) The circadian period length of all tested genotypes (wild-type: n = 18, *Opn4^LacZ/LacZ^*: n = 17, *Opn4^DTA/+^*: n = 8, *Opn4^DTA/LacZ^*: n = 7, *Opn4^DTA/DTA^*: n = 7, and *Opn4^aDTA/aDTA^*: n = 18). (**C**) For phase-shifting experiments, a subset of wild-type, *Opn4^LacZ/LacZ^*, *Opn4^aDTA/aDTA^* were assessed, and the total mice analyzed was as follows: wild-type (n = 7), *Opn4^LacZ/LacZ^* (n = 8), *Opn4^DTA/+^* (n = 8), *Opn4^DTA/LacZ^* (n = 7), *Opn4^DTA/DTA^* (n = 7), and *Opn4^aDTA/aDTA^* (n = 8). As expected from substantial ipRGC ablation, phase shifting was significantly reduced in *Opn4^DTA/LacZ^*, *Opn4^DTA/DTA^* and *Opn4^aDTA/aDTA^* mice. (**D**) Representative actograms of wild-type mice enucleated at either P0 (n = 8) or P60 (n = 7), under 12:12-LD and DD. (**E**) The circadian period length for both enucleation groups. Only mice enucleated at P0 exhibited a lengthened circadian period (**B and C**) One-way ANOVA with Bonferroni's multiple comparisons test and adjusted p values. (**E**) Two-way ANOVA, Bonferroni's multiple comparisons test and adjusted p values. Error bars represent s.e.m. for all graphs. See also *Figure 2—figure supplements 1–2*.

*Figure 2 continued on next page*

*Figure 2 continued*

The following figure supplements are available for figure 2:

**Figure supplement 1.** Additional actograms for *Opn4^{DTA/LacZ}* and *Opn4^{DTA/DTA}* mice.

**Figure supplement 2.** Expression pattern of transcription factors critical for SCN development.

early development, these findings suggested the hypothesis that ipRGCs act during early postnatal ages to establish a normal circadian period length. We performed a general examination of expression of transcription factors important for SCN development and a neuropeptide relevant for clock function to examine the SCN, and found no apparent disruption in the SCN of *Opn4^{DTA/DTA}* mice (*Figure 2—figure supplement 2*). Thus, loss of ipRGC innervation does not drastically alter SCN development, and it is possible the lengthened period stems from a loss of direct ipRGC-dependent regulation of SCN neurons. Alternatively, this effect could be due to altered input from the intergeniculate leaflet (IGL), which also receives innervation by ipRGCs.

To assess the relevance of early ablation of ipRGCs in causing the lengthened circadian period of *Opn4^{DTA/DTA}* mice, we removed both eyes from mice at either P0 or P60 to mimic the killing of ipRGCs early (*Opn4^{DTA/DTA}*) versus late (*Opn4^{aDTA/aDTA}*). Starting at P74, we recorded wheel-running activity under a 12:12-LD cycle and constant darkness. Since the mice lack eyes, they lacked photic effects on the clock (*Figure 2D,E*). However, mice enucleated at P0 exhibited a lengthened circadian period similar to *Opn4^{DTA/DTA}* mice, whereas mice enucleated at P60 phenocopied *Opn4^{aDTA/aDTA}* mice exhibiting a period comparable to intact mice (*Figure 2A–B,D–E*). Circadian period length has been shown to be stable in adult mice kept under constant darkness for 2 months (*Campuzano et al., 1999*). Thus, these data corroborate results from *Opn4^{DTA/DTA}* mice indicating that ipRGCs function postnatally to set the length of the circadian period.

## Light is required to set the length of the intrinsic circadian period

We asked whether light-driven input from ipRGCs is important for establishing circadian period length. Wild-type animals were raised under either constant darkness or a 12:12-LD cycle. At P60, we assessed the intrinsic circadian period of these animals by recording their wheel-running activity in constant darkness (*Figure 3A,B*; *Figure 3—figure supplement 1*). While the phenotype was less penetrant than that the *Opn4^{DTA/DTA}* mice and P0 enucleates, most (9 of 16) dark-reared mice exhibited a longer intrinsic circadian period than mice raised in a 12:12-LD cycle and this was stable for duration of our recordings (up to 60 days; *Figure 3A,B*; *Figure 3—figure supplement 1*). The lengthened period of the dark-reared animals was comparable to that exhibited by mice enucleated at P0 and mice with early postnatal ablation of ipRGCs (*Opn4^{DTA/DTA}*) (*Figures 2* and *3*; *Figure 3—figure supplement 1*). It has been thought that the mammalian circadian period length was established independent of sensory input, but our data show that light contributes to this process. In zebrafish, it is known that light is required to initiate expression of clock genes and for generation of rhythms (*Ben-Moshe et al., 2014*; *Kazimi and Cahill, 1999*; *Ziv et al., 2005*); however, this is the first evidence to indicate that light is also required for mammalian clock maturation.

Upon exposure to a 12:12-LD-cycle, dark-reared mice photoentrained, and light exposure during photoentrainment was sufficient to normalize the circadian period length (*Figure 3A,B*; *Figure 3—figure supplement 1*). Furthermore, a single short light pulse (3 hr), which did not cause photoentrainment, (*Figure 3C,D*) was also sufficient to set the circadian period of dark-reared animals (*Figure 3C,D*). These data indicate that brief light exposure is sufficient to set the circadian period and can even occur in adults, thus, indicating a lack of a critical developmental window for this process.

## Developmental ablation of ipRGCs disrupts axonal segregation of retinogeniculate projections

By P7, ipRGCs have reached their central targets in the brain (*Figure 1A–B*) (*McNeill et al., 2011*) and a subset of ipRGCs forms intra-retinal collateral axons that terminate in the inner plexiform layer,

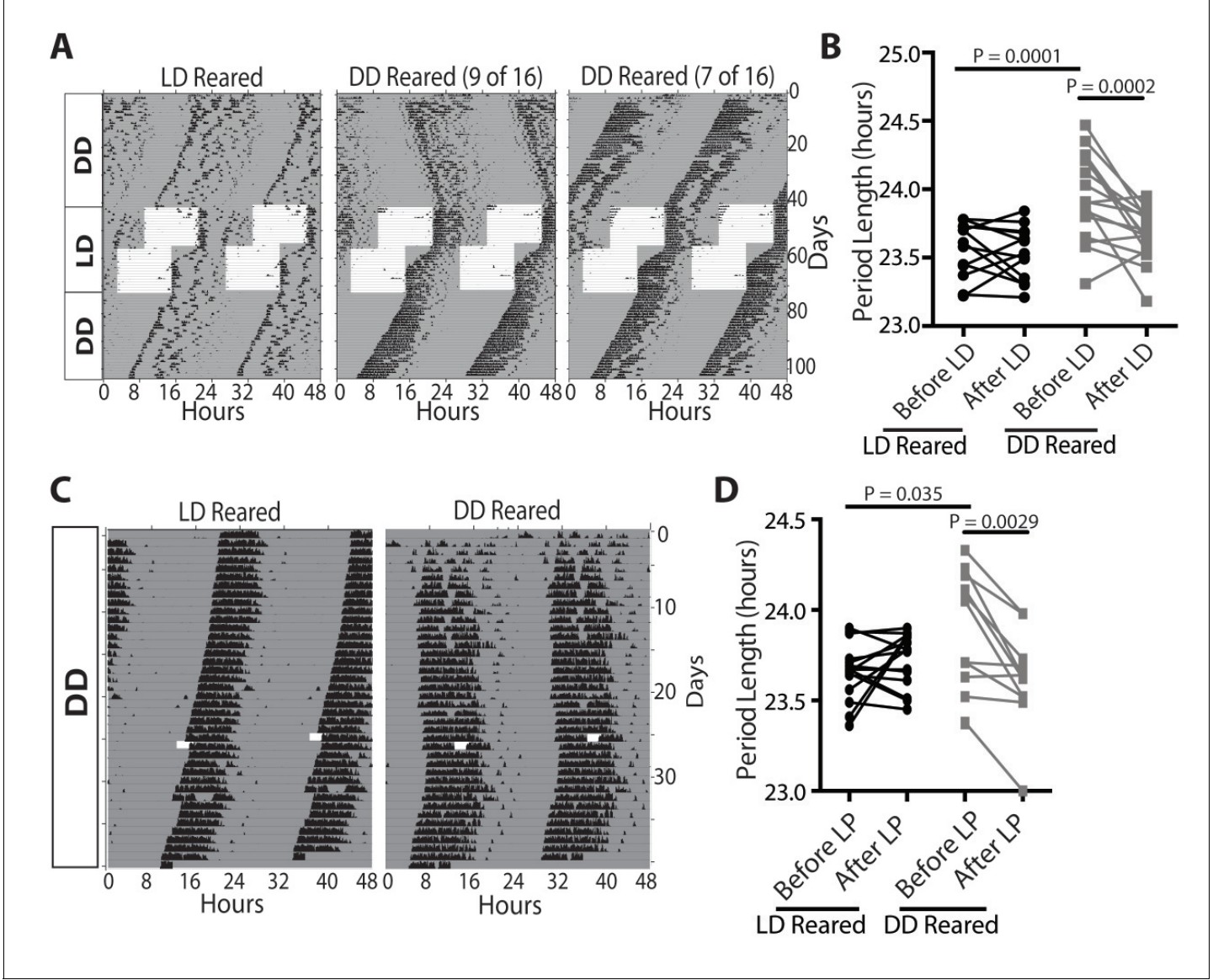

**Figure 3.** Light sets the circadian period length, even during adulthood. (**A and B**) Representative actograms of wild-type mice raised in either a 12:12-LD cycle (n = 13) or darkness (n = 16), and exposed to 12:12-LD cycle for 1 month. The circadian period of dark-reared mice is lengthened initially, but rescued and is no longer significantly different from 12:12-LD reared mice after light exposure. (**C and D**) Representative actograms of wild-type mice raised in either a 12:12-LD cycle (n = 17) or in darkness (n = 12), and then exposed to a 3 hr light pulse. Dark-reared mice exhibited a longer period. Following the 3 hr light pulse, the circadian period length of dark-reared animals shortened. (**B and D**) Two-way ANOVA, Bonferroni's multiple comparisons test and adjusted p values. Error bars represent s.e.m. See also *Figure 3—figure supplement 1*.

The following figure supplement is available for figure 3:

**Figure supplement 1.** Actograms for all dark reared mice tested.

where they synapse onto dopaminergic amacrine cells (*Joo et al., 2013*; *Prigge et al., 2016*) (*Figure 4—figure supplement 1*). P7 is in the middle of a critical developmental period for the image-forming visual system. From P0-P14, the retinotopic map is being refined by spontaneously generated neural activity in the retina, termed retinal waves (*Ackman et al., 2012*; *Feller, 2009*; *Feller et al., 1996*; *McLaughlin et al., 2003*; *Meister et al., 1991*; *Ramoa et al., 1989*; *Shatz, 1990*; *Stellwagen and Shatz, 2002*; *Xu et al. 2011*; *Zhang et al., 2011*). The presence of intra-retinal collaterals on ipRGCs at early postnatal ages provides anatomical means for ipRGCs to influence the

retina during an important developmental window. The $Opn4^{DTA/DTA}$ mice allowed us the opportunity to directly test the involvement of ipRGCs, not just melanopsin-based light sensitivity, in the development and refinement of the image-forming visual system. To investigate this, we traced the axonal projections of retinal ganglion cells in adult wild-type and $Opn4^{DTA/DTA}$ mice, and observed that, while retinal innervation of the superior colliculus in $Opn4^{DTA/DTA}$ mice was comparable to wild-type (*Figure 4—figure supplement 2*), eye-specific axonal segregation in the dLGN was severely disrupted (*Figure 4A,B*; *Figure 4—figure supplements 3* and *4*). In $Opn4^{DTA/DTA}$ mice, the ipsilateral zone had poorly defined boundaries and substantial contralateral innervation (*Figure 4A*). We quantified eye-specific axonal segregation in the dLGN using two distinct methods (*Datwani et al., 2009*; *Demas et al., 2006*; *Renna et al., 2011*) (*Figure 4B*; *Figure 4—figure supplement 3*). In $Opn4^{DTA/DTA}$ mice, there was a significant increase in the percentage of pixels with overlapping input from the two eyes, and this eye-specific segregation defect was most severe in the caudal dLGN (*Figure 4A*, *Figure 4—figure supplement 4F*). The total amount of ipsilateral and contralateral input and total dLGN size were similar across all tested genotypes (*Figure 4C,D*, *Figure 4—figure supplement 4C–E*). Abnormal eye-specific segregation was not observed in either melanopsin knockout mice ($Opn4^{LacZ/LacZ}$) or mice with ipRGCs ablated during adulthood ($Opn4^{aDTA/aDTA}$) (*Figure 4—figure supplement 4A–D*). Importantly, $Opn4^{DTA/+}$ mice, which have substantial ipRGC loss, although to lesser degree than $Opn4^{DTA/DTA}$ mice, exhibited disrupted eye-specific segregation that was intermediate between wild-type and $Opn4^{DTA/DTA}$ mice (*Figure 4A,B*, *Figure 4—figure supplement 4F*). Combined, these data reveal an developmental role for ipRGCs in the segregation of all RGC inputs to the dLGN.

To conclusively determine whether ipRGCs function during early postnatal ages to regulate refinement of retinogeniculate projections, we examined eye-specific segregation at P8 and, similar to adult mice, found an increase in overlapping ipsilateral and contralateral projections in $Opn4^{DTA/DTA}$ mice, but no difference in total retinal input compared to wild-type (*Figure 5A–D*). These data indicate that ipRGCs function at early postnatal ages to mediate refinement of the image-forming visual system by regulating eye-specific segregation of retinogeniculate projections.

## Mice with early ablation of ipRGCs have deficits in visual acuity

To examine the functional consequence of altered eye-specific segregation, we compared wild-type, $Opn4^{LacZ/LacZ}$, and $Opn4^{DTA/DTA}$ mice in two behavioral tests of visual acuity: the virtual optomotor system and the visual water task (*Douglas et al., 2005*; *Prusky et al., 2000*). By both measures, $Opn4^{DTA/DTA}$ animals exhibited reduced visual acuity compared to controls (wild-type and $Opn4^{LacZ/LacZ}$; *Figure 4E,F*).

Heterozygous DTA animals ($Opn4^{DTA/+}$) have less ipRGC loss than homozygous mice ($Opn4^{DTA/DTA}$) (*Figure 1D*; *Supplementary file 1*) and have intermediate deficits in eye-specific segregation compared to homozygous and wild-type mice (*Figure 4A,B*; *Figure 4—figure supplement 4F*). Remarkably, $Opn4^{DTA/+}$ mice also exhibited an intermediate reduction in visual acuity (*Figure 4E,F*). Thus, the severity of the deficits in eye-specific segregation and visual functions were correlated with the extent of ipRGC loss. While many factors including circadian time and the pupillary light reflex contribute to visual acuity, these factors cannot explain the deficits in visual acuity observed in $Opn4^{DTA/+}$ and $Opn4^{DTA/DTA}$ mice. First, the loss of acuity in $Opn4^{DTA/DTA}$ cannot be explained by a loss of photoentrainment and the possibility that we are testing acuity at different circadian times, because $Opn4^{DTA/+}$ mice, which photoentrain also exhibit deficits in visual acuity. In addition, $Opn4^{aDTA/aDTA}$ mice, which free run, do not exhibit the substantial reduction in acuity observed in $Opn4^{DTA/DTA}$ mice. Furthermore, loss of the pupillary light reflex cannot fully explain the deficits in visual acuity because when the pupil is fully dilated with atropine there is only a minor reduction in acuity and it is not comparable to the substantial deficits observed in $Opn4^{DTA/+}$ and $Opn4^{DTA/DTA}$ mice (*Güler et al., 2008*). However, it is important to note that we cannot rule out contributions from direct effects of ipRGCs on retinal functions. For example, the role of M1 ipRGCs in regulating dopamine in the retina, could contribute (*Dkhissi-Benyahya et al., 2013*; *Prigge et al., 2016*; *Zhang et al., 2012*).

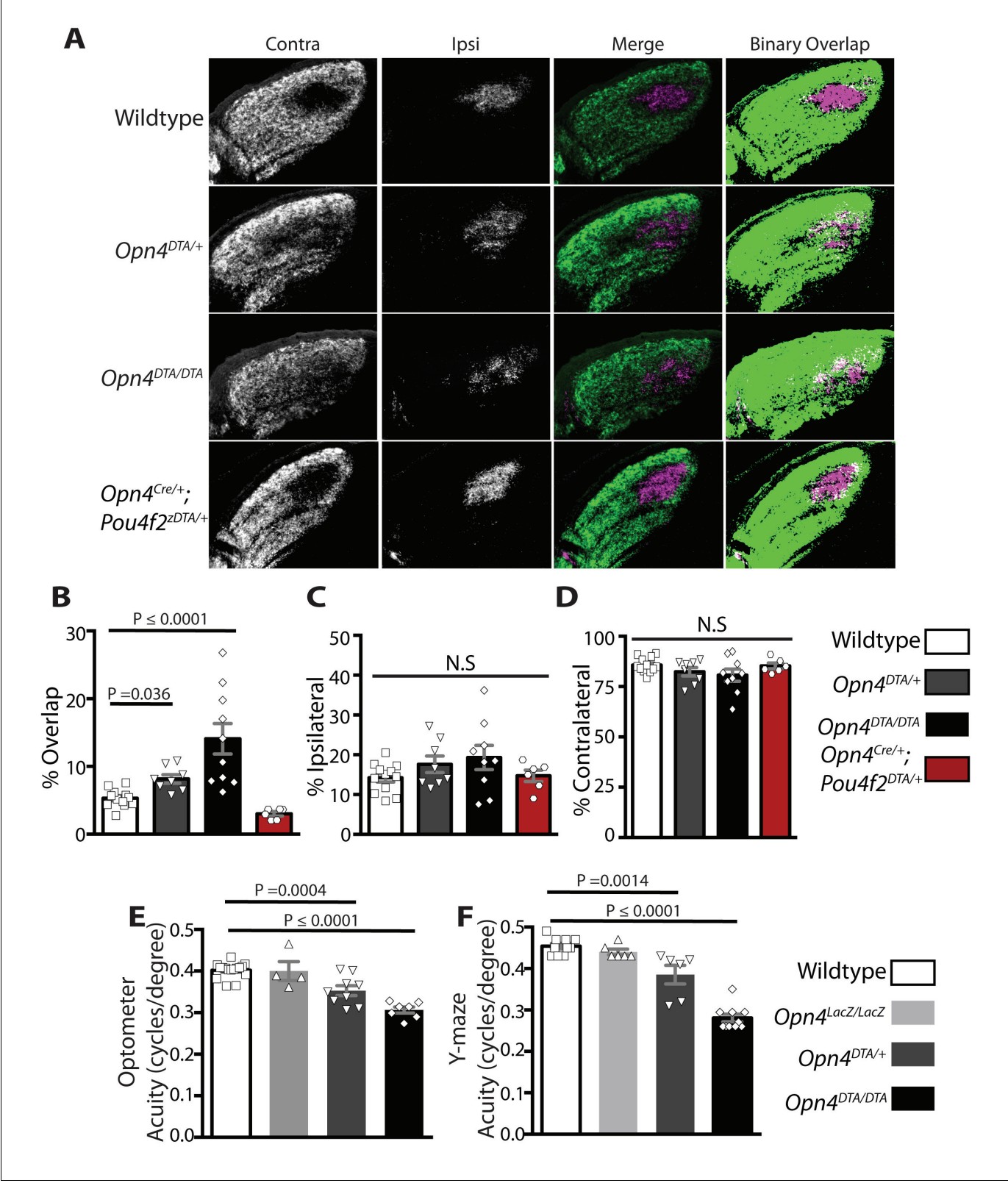

**Figure 4.** Adult *Opn4^DTA/DTA* mice display deficits of eye-specific axonal segregation and visual acuity. (**A**) RGC axonal innervation of the adult dLGN of wild-type (n = 12), *Opn4^DTA/+* (n = 8), *Opn4^DTA/DTA* (n = 10) and *Opn4^Cre/+; Pou4f2^zDTA/+* (n = 6; previously published as *Opn4^Cre/+; Brn3b^zDTA/+*). The rightmost images represent binarized version of the merged images to visualize the overlap between contralateral and ipsilateral RGC projections. Representative images were taken from the region of the dLGN indicated by the blue arrow in *Figure 4—figure supplement 4F*. (**B**) *Opn4^DTA/DTA* mice

*Figure 4 continued on next page*

*Figure 4 continued*

exhibited a significantly higher percentage of overlapping pixels relative to the total number of LGN than any other tested genotype. $Opn4^{DTA/+}$ mice exhibited levels of overlapping pixels that were intermediate compared to $Opn4^{DTA/DTA}$ and control mice. (**C and D**) The percentage of the total number of pixels in the dLGN from ipsilateral and contralateral fibers is similar among all tested genotypes. (**E and F**) The virtual optokinetic system and visual water task were used to assess visual function wild-type (virtual optokinetic system: n = 16, visual water task: n = 10), $Opn4^{LacZ/LacZ}$ (virtual optokinetic system: n = 4, visual water task: n = 6), $Opn4^{DTA/+}$ (virtual optokinetic system: n = 9, visual water task: n = 6), $Opn4^{DTA/DTA}$ (virtual optokinetic system: n = 7, visual water task: n = 10) mice. Wild-type and $Opn4^{LacZ/LacZ}$ mice were indistinguishable. $Opn4^{DTA/DTA}$ mice exhibited reduced visual acuity compared to wild-type mice, and $Opn4^{DTA/+}$ mice exhibited intermediate visual acuity. (**B–F**) One-way ANOVA, Bonferroni's multiple comparisons test and adjusted p values. Error bars represent s.e.m. See also *Figure 4—figure supplements 1–5*.

The following figure supplements are available for figure 4:

**Figure supplement 1.** ipRGC intra-retinal axonal collaterals are present by P7.

**Figure supplement 2.** Retinal innervation of the SC in $Opn4^{DTA/DTA}$ mice is indistinguishable from wild-type mice.

**Figure supplement 3.** Quantification of disruption in eye-specific axonal segregation in the dLGN of adult $Opn4^{DTA/DTA}$ mice.

**Figure supplement 4.** Eye-specific axonal segregation is normal in $Opn4^{Cre/+}$, $Opn4^{LacZ/LacZ}$, and $Opn4^{aDTA/aDTA}$ mice.

**Figure supplement 5.** All but ~200 ipRGCs are ablated by P7 in $Opn4^{Cre/+}$; $Pou4f2^{zDTA/+}$ mice.

## Spontaneous retinal activity is altered in $Opn4^{DTA/DTA}$ mice

Eye-specific segregation deficits in $Opn4^{DTA/DTA}$ mice were observed by P8, and since light through ipRGCs modulates retinal wave activity and retinal waves drive ipRGC spiking (*Renna et al., 2011*), we hypothesized that $Opn4^{DTA/DTA}$ have altered spontaneous retinal activity in darkness. We recorded wave activity at P6 in the dark on a multielectrode array in wild-type and $Opn4^{DTA/DTA}$ retinas (*Figure 5E–H*; *Supplementary file 2*). Spiking properties of RGCs during waves were significantly altered in $Opn4^{DTA/DTA}$ mice. Wave-associated bursts (WABs) were significantly longer in duration, and had higher firing rates, and shorter inter-burst intervals (*Figure 5E,G–H*, *Supplementary file 2*) than they did in wild-type controls. There were also significantly more total spikes as well as more spikes outside of bursts (*Figure 5E,G–H*, *Supplementary file 2*). Correlated spiking activity between neurons was very similar between the genotypes, as measured by the spike time tiling coefficient (STTC; [*Cutts and Eglen, 2014*])(*Figure 5F*). These data indicate that ipRGCs are critical for normal retinal wave activity, even in darkness, and together with our anatomical findings (*Figure 4A–D*; *Figure 4—figure supplements 3–4*), allow us to suggest that ipRGCs mediate eye-specific segregation of RGC projections to the dLGN by regulating the spiking properties of conventional RGCs during retinal waves.

## 200 M1 ipRGCs are sufficient for both setting the circadian period and refining the imaging-forming visual system

There are currently five identified subtypes of ipRGCs. M1 ipRGCs project to non-image-forming brain centers, while M2-M5 ipRGCs project at least in part to image-forming targets (*Ecker et al., 2010*; *Schmidt et al., 2011*; *Schmidt and Kofuji, 2009*). The M1 subtype can be further subdivided based on expression of the transcription factor *Pou4f2* (also referred to as Brn3b), which is also expressed in all non-M1 ipRGCs (*Chen et al., 2011*).~200 *Pou4f2*-negative-M1 ipRGCs project exclusively to circadian centers (predominantly to the SCN) (*Figure 4—figure supplement 5B*) (*Chen et al., 2011*) and are the subset of ipRGCs that have intra-retinal collateral axons (data not shown). This population of cells are sufficient for circadian photoentrainment in the presence of one copy of melanopsin (*Chen et al., 2011*). As described previously, *Pou4f2*-positive M1 ipRGCs and non-M1 ipRGCs can be selectively ablated by crossing $Opn4^{Cre/+}$ mice with $Pou4f2^{Z-DTA/+}$ mice (previously published as $Brn3b^{Z-DTA/+}$ mice), in which a floxed stop cassette followed by DTA was inserted into the *Pou4f2* locus (*Chen et al., 2011*; *Mu et al., 2005*). In doubly heterozygous offspring ($Opn4^{Cre/+}$; $Pou4f2^{Z-DTA/+}$, previously published as $Opn4^{Cre/+}$;$Brn3b^{Z-DTA/+}$ mice, *Supplementary file 1*), *Pou4f2*-positive ipRGCs (some M1 and all non-M1 ipRGCs) are ablated by P7, leaving a subset of

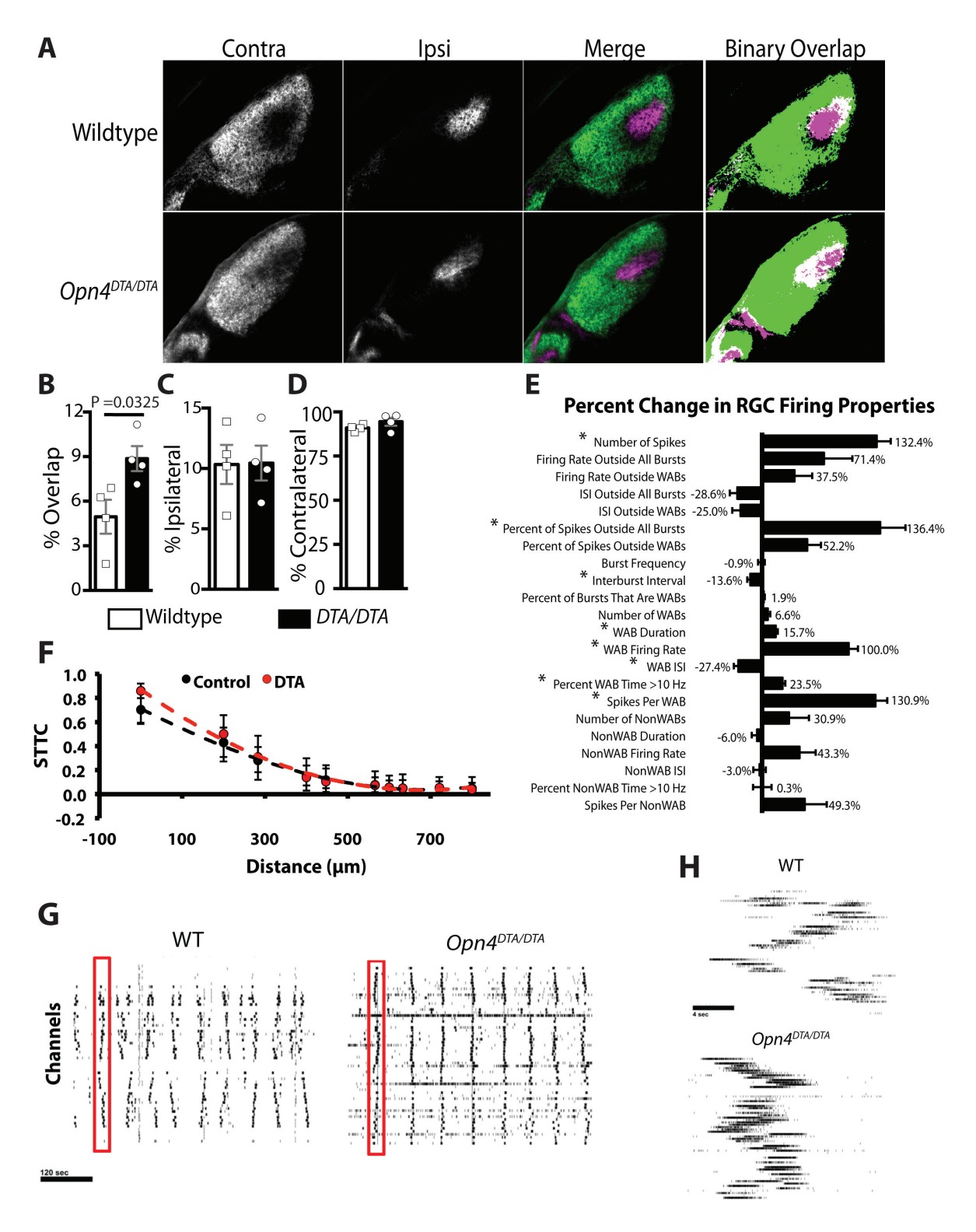

**Figure 5.** *Opn4DTA/DTA* mice display disrupted eye-specific axonal segregation as early as P8 and exhibit altered RGC firing properties. (**A**) RGC axonal innervation of the dLGN in P8 wild-type (n = 4) and *Opn4DTA/DTA* (n = 4) mice as in *Figure 3A*. (**B**) At P8, *Opn4DTA/DTA* mice exhibit a significantly higher percentage of overlapping pixels relative to the total number of LGN pixels than wild-type mice. (**C and D**) The percentage of the total number of pixels in the dLGN from ipsilateral and contralateral fibers is similar among all tested genotypes. * indicates p<0.05 with Student's t-test. Error bars
*Figure 5 continued on next page*

*Figure 5 continued*

represent s.e.m. for all graphs. (**E**) RGC spiking properties. *Opn4^DTA/DTA* mice exhibit longer WABs and had a higher firing rate, with more spikes and a shorter inter-burst interval. There were also significantly more spikes outside of WABs. (**F**) Spike Time Tiling Coefficient (STTC) versus distance demonstrating that correlated spiking activity between neurons was similar in *Opn4^DTA/DTA* mice compared to controls. 275 spike trains from wild-type/control, 245 from DTA. * represent statistically significant differences after Student's T-tests with a Holm-Bonferroni correction, α = 0.05, m = 22 for spiking properties, m = 10 for STTC.' Exact p values are listed in ***Supplementary file 2***. (**G**) Representative raster plots from multielectrode array recording of retinal waves in P6 WT and *Opn4^DTA/DTA* mice in the dark. Each row represents the activity on a single electrode. (**H**) Expansion of one wave (identified by the red box in (**G**)). Error bars represent s.e.m. for all graphs.

M1 *Pou4f2*-negative ipRGCs (~200 cells) and all conventional RGCs (***Chen et al., 2011***) (***Figure 4—figure supplement 5A***, ***Supplementary file 1***). In these mice, the SCN remains innervated by ipRGCs, the IGL receives partial innervation, and ipRGC innervation of image-forming brain regions is entirely abolished (***Chen et al., 2011***) (***Figure 4—figure supplement 5B***).

In *Opn4^Cre/+*; *Pou4f2^Z-DTA/+* mice, ablation of *Pou4f2*-positive ipRGCs is complete by P7, and at this age only the ~200 *Pou4f2*-negative ipRGCs remain (***Figure 4—figure supplement 5A***; ***Supplementary file 1***). *Opn4^Cre/+*; *Pou4f2^Z-DTA/+* mice can photoentrain and free-run with a circadian period indistinguishable from controls (***Chen et al., 2011***), indicating that the remaining *Pou4f2*-negative-M1 ipRGCs are sufficient to set the circadian period. Organization of eye-specific retinogeniculate projections in *Opn4^Cre/+*; *Pou4f2^Z-DTA/+* is also indistinguishable from controls (***Figure 4A–D***; ***Figure 4—figure supplement 4A–D***). Together these results demonstrate that a single subpopulation of ipRGCs are critically involved in the development of networks devoted to image-forming vision and the circadian clock. This small population of cells is sufficient for setting of the circadian period as well as proper segregation of retinogeniculate circuitry.

## Discussion

In this study, we identified a shared developmental mechanism for the maturation of the circadian clock and refinement of eye-specific segregation. Our work reveals the unprecedented role that light plays in setting the circadian period and identified a regulatory component of the neural networks that regulate refinement of RGC projections to the thalamus.

The methods we used to ablate ipRGCs involve some inherent uncertainties. We used two different versions of diphtheria toxin subunit A with different potencies. As expected, the full-strength DTA was more efficient than was the attenuated form at killing ipRGCs, but it also ablated ipRGCs earlier in development. One concern of using two copies of the toxin gene is that even low leaky expression could lead to off-target cell death. We think this is unlikely since the toxin failed to ablate at least one ipRGC subtypes with low levels of melanopsin expression—the M4 cells. Moreover, loss of cells in the retina and retinofugal projections in the brain were consistent with selective loss of ipRGCs. However, it is always possible that DTA expression from the melanopsin locus causes a reduction in cell populations that we did not examine. Further, because DTA blocks protein translation, it could disrupt the function of some cells without actually killing them; the assays we performed do not distinguish between these forms of toxin action. In addition, technical limitations precluded full assessment the degree of cell loss for many ipRGC subtypes. We currently lack specific molecular markers for M2, M3, and M5 cells. Furthermore, dendritic stratification is key to distinguishing ipRGC subtypes, but DTA-dependent ipRGC loss occurs before dendritic arborizations are mature (between P10-P15, ***Coombs et al., 2007***). However, it is clear that, in *Opn4^Cre/+*; *Pou4f2^Z-DTA/+* mice, *Pou4f2*-positive ipRGCs, which include all non-M1 ipRGCs, die early during development and yet, their death does not affect the circadian period or the segregation of RGC projections in the dLGN. Thus, despite some uncertainty about which ipRGCs types die in these mouse lines and when they do so, taken together our data support the view that early ablation of *Pou4f2*-negative M1 ipRGCs are sufficient to induce the effects reported here.

Our study indicates that contrary to the prevailing view that the circadian clock develops independent of environmental input, light is necessary for setting the intrinsic period of the circadian clock. When wild-type animals were raised in constant darkness, they exhibited a lengthened circadian period as was observed when mice were enucleated at P0 and when ipRGCs were ablated at early

postnatal ages ($Opn4^{DTA/DTA}$) but not when mice were enucleated at P60 or when ipRGCs were only ablated during adulthood ($Opn4^{aDTA/aDTA}$; *Figures 2* and *3*). Remarkably, when dark-reared mice were exposed to light for the first time during adulthood, their period length became indistinguishable from mice reared under a 12:12 LD cycle (*Figure 3*). This rescue indicates there is no critical developmental window during which the intrinsic properties of the circadian clock must be set by light. Moreover, once animals are exposed to light, the circadian period is irreversibly set. It will be of interest to investigate the molecular and cellular mechanisms underlying the lengthened period in $Opn4^{DTA/DTA}$, P0 enucleates, and dark reared mice, and how light detection by ipRGCs induces a permanent change in the period length of the circadian clock.

We noticed that the phenotype exhibited by dark-reared animals was less penetrant than that of $Opn4^{DTA/DTA}$ and P0 enucleates. Though a majority of dark-reared animals (9 of 16) exhibited a lengthened period, the remainder did not (*Figure 3—figure supplement 1*). By contrast, virtually all $Opn4^{DTA/DTA}$ mice and P0 enucleates exhibited lengthened circadian periods. One possible interpretation of this variability could be that the process of setting the circadian period is highly sensitive to light. This is supported by the fact that merely 3 hr of light exposure is sufficient to set the circadian period (*Figure 3C,D*). Alternatively, some dark-reared animals could have been exposed to, say, very weak light occasionally leaking from night vision goggles or it is possible that, since ipRGCs depolarize during retinal waves, spontaneously derived activity can partially suffice for period setting. It would be interesting, in future studies, to dark rear mice that lack retinal waves and determine whether a higher proportion exhibit a lengthened period than dark-reared wild-type mice.

A surprising finding was that early genetic ablation of ipRGCs led to disrupted eye-specific segregation in the dLGN and reduced visual acuity (*Figures 4* and *5*; *Figure 4—figure supplements 3* and *4*). Even more surprisingly, our data implicated *Pou4f2*-negative M1 ipRGCs, which have been viewed as circadian photoreceptors, as being involved in the development of the image-forming visual system (*Figure 4A,B*; *Figure 4—figure supplement 4A–B*). In $Opn4^{Cre/+}$; $Pou4f2^{Z-DTA/+}$ mice, ipRGC subtypes known to innervate the dLGN (i.e., M2-M5 cells) are developmentally ablated (*Figure 4—figure supplement 5*) and only ~200 *Pou4f2*-negative, circadian center projecting-M1 ipRGCs remain (*Chen et al., 2011*). In these animals, geniculate organization and visual acuity are normal (*Figure 4A–D*) (*Chen et al., 2011*). The further loss of these 200 *Pou4f2*-negative M1 ipRGCs in $Opn4^{DTA/DTA}$ mice (*Figure 1*) resulted in severe deficits in image-forming visual system (*Figures 4* and *5*; *Figure 4—figure supplements 3* and *4*). These results also suggested that the defects observed in $Opn4^{DTA/DTA}$ mice are not due to the loss of direct ipRGC innervation of the dLGN, nor to a generalized reduction in the total number of RGCs.

It is possible that the segregation deficits observed in $Opn4^{DTA}$ mice are due to the altered spontaneous retinal activity that occurs in P6 $Opn4^{DTA/DTA}$ mice. The salient features of retinal waves appear to be comprised of correlated, patterned activity sweeping across the retina, instructive for the formation of retinofugal circuits in the superior colliculus (*Xu et al. 2015*; *Xu et al. 2011*), whereas the individual spiking properties of ganglion cells drive eye-specific segregation within the dorsal lateral geniculate nucleus (*Speer et al., 2014*). Many previous studies have examined mice with genetically or pharmacologically disrupted spontaneous activity (*Blankenship et al., 2011*; *Rossi et al., 2001*; *Stellwagen et al., 1999*; *Torborg et al., 2005*; *Xu et al. 2011*), yet, $Opn4^{DTA/DTA}$ mice are a unique instance of a manipulation that caused an increase in non-WAB firing while WAB activity remains highly correlated between pairs of RGCs. Connexin36/45 double knockouts, which have disrupted eye-specific segregation, exhibit increased tonic firing in RGCs, although correlated firing between RGCs is also highly reduced (*Blankenship et al., 2011*). In ferret, disruption of patterned activity by the ablation of starburst amacrine cells altered correlated activity and a number of WAB bursting properties, yet eye-specific segregation proceeded normally (*Speer et al., 2014*). Our data are consistent with this study, suggesting changes in spiking activity outside of WABs is critical for eye specific sectors within the dLGN.

The contribution of disrupted eye-specific segregation to reduced visual acuity has yet to be thoroughly investigated likely because many mutations or manipulations that cause disrupted eye-specific segregation also disrupt other functions of the visual system. For example, no b-wave (*nob*) mice have disrupted eye-specific segregation (*Demas et al., 2006*) and have behavioral deficits in visual acuity, but this more likely explained by their substantially altered ERG responses (*Neuillé et al., 2014*). Similarly eye injections of TTX or epibatidine would disrupt acuity for reasons other than disrupting eye-specific segregation. Nonetheless, *β2 KO* mice, which have eye-specific

segregation deficits at P8 but not at P28 (*Feller, 2002*), have a substantial deficit in visual acuity in the optomotor response and by recordings of visually evoked potentials (*Rossi et al., 2001*), but exhibit no deficit in the Y-maze (*Wang et al., 2009*). Thus, it is plausible that the disrupted eye-specific segregation in $Opn4^{DTA/DTA}$ and $Opn4^{DTA/+}$ causes the reduction in visual acuity, but as discussed above, we cannot rule out the contribution of other roles for ipRGCs in retina.

*Pou4f2*-negative M1 ipRGCs project intra-retinal axonal collaterals that synapse onto dopaminergic amacrine cells, and these projections are present by P7 (*Prigge et al., 2016*) (*Figure 4—figure supplement 1*). Thus, it is probable that ipRGCs regulate the spiking properties of RGCs through intra-retinal signaling via axonal collaterals.

In this study, we found that maturation of the circadian clock requires light input and setting of the circadian period can occur even in adulthood. We also showed that ipRGCs are necessary for refinement of eye-specific axonal segregation in the LGN as well as normal visual acuity. Our work reveals that while the networks underlying the image-forming and NIF systems have long been viewed as distinct, ipRGCs constitute a shared node in the neural circuits that mediate light-dependent maturation of the circadian clock and refinement of retinogeniculate projections.

## Materials and methods

### Animal models
Animals were housed and treated in accordance with NIH and IACUC guidelines, and used protocols approved by the Johns Hopkins University and Brown University Animal Care and Use Committees (Protocol numbers MO16A212, and 1010040).

### Statistical analysis
All statistical tests were performed in Graphpad Prism 6, except for retinal wave analysis the detail of which are described below. Specific statistical comparisons are listed in the figure captions.

### Generation of $Opn4^{DTA}$ mice
To generate $Opn4^{DTA}$ mice, we used the targeting arms and general strategy detailed in (*Ecker et al., 2010*; *Güler et al., 2008*; *Hattar et al., 2002*). The construct contained a 4.4 kb sequence immediately 5' of the start codon for mouse melanopsin, followed by the coding sequence for diphtheria toxin A (DTA) subunit, an internal ribosomal entry site (IRES), the coding sequence for tauLacZ, and a self-excising neomycin resistance construct (loxP-tAce-Cre-Pol II-Neo-loxP) (*Figure 1—figure supplement 1A*). Embryonic stem (ES) cells were first screened for homologous recombination by PCR, and then homologous recombination was confirmed with Southern blot analysis (a restriction digestion with SpeI resulted in a 10.7 kb band for the wild-type allele and a 5.3 kb band for the recombined allele) (*Figure 1—figure supplement 1B*). The blastocyst injection was performed by the Johns Hopkins transgenic core facility. The germline transmission was obtained by crossing chimeric males with C57Bl/6J females. The genotyping was done by PCR. The DTA allele was detected with the primers: AACTTTTCTTCGTACCACGG (forward) and ACTCATACATCGCATC TTGG (reverse), and the wild-type allele was detected with the primers: CCCCTGCTCATCATCATC TTCTG (forward) and TGACAATCAGTGCGACCTTGGC (reverse). $Opn4^{DTA/+}$ and $Opn4^{DTA/DTA}$ mice are viable, fertile, and do not exhibit any gross abnormalities in size.

### Alkaline phosphatase staining
A cre-mediated alkaline phosphatase (AP) reporter, provided by Tudor Badea in Jeremy Nathan's lab, was expressed in conjunction with $Opn4^{Cre}$ (*Ecker et al., 2010*). Mice were deeply anesthetized with 30 ml/kg Avertin and then intracardially perfused with phosphate-buffered saline for 3 min followed by 40 ml of 4% paraformaldehyde. Brains and retinas were isolated and post-fixed for 40 min in 4% paraformaldehyde. Brains were mounted in 3% agarose and then cut into 200 µm sections on a vibrating microtome (Vibra-tome 1000 Plus). Tissue was heat-inactivated for overnight at 65°C. Alkaline phosphatase histochemistry was performed using NBT/BCIP tablets (Roche) for 2–4 hr in the dark with constant shaking. Tissue was washed three times with phosphate-buffered saline containing 0.1% Tween-20 (Sigma-Aldrich). Retinas were mounted immediately and imaged. Brains were fixed 3 hr in 4% paraformaldehyde at 4°C, then counterstained with 1:5 Fast Red nuclear stain

(Vector Laboratories) in water for 7 min. The sections were then dehydrated in an ethanol series, and after at least in hour in 100% ethanol, the sections were cleared in a 2:1 mixture of benzyl benzoate: benzyl alcohol (Sigma-Aldrich), mounted in glycerol, and imaged immediately. To measure cell density, we counted the number of ipRGCs, in four representative areas of each retina, and calculated the density of ipRGCs per $mm^2$.

## X-gal staining

Mice were deeply anesthetized with 30 ml/kg Avertin followed by cervical dislocation. Eyes were isolated and fixed in 4% paraformaldehyde for 10 min. Retinas were dissected out and then incubated in buffer B (100 mM phosphate buffer at pH 7.4, 2 mM $MgCl_2$, 0.01% sodium deoxycholate, 0.02% IGEPAL) then stained for 3 days in buffer B plus 5 mM potassium ferricyanide, 5 mM potassium ferrocyanide and 1 mg/ml X-gal as described in *Hattar et al. (2002)*.

## Hematoxilin and eosin staining

Animals were anesthetized with 30 ml/kg Avertin, and eyes were removed fixed in 4% PFA for 1 hr. Retinas were dissected in PBS, placed in cartridges (Tissue-Tek Biopsy Uni-Cassette), and processed and embedded in paraffin overnight. Eyecups were sectioned at 6 μm. Resulting sections were deparaffinize by immersion in two changes of xylene for 10 min each. Sections were then rehydrated in descending series of ethanol ending in water for 5 min. Sections were stained with hematoxylin for 30 s, washed with tap water for 10 dips, placed briefly in 0.1% sodium bicarbonate, and then rinsed in clean tap water for 10 dips. Sections were rinsed in 70% ethanol for five dips and stained in eosin for 20 s. Sectioned were dehydrated with an ascending series of ethanol, ending with two washes of 100% ethanol. Sections were placed in two washes of xylene (5 min each), and mounted in Permount.

## Immunohistochemistry

Whole eyes were fixed for 15 min in 2% paraformaldehyde (PFA) diluted in PBS and were then dissected to remove the cornea and lens. Whole eyecups were fixed for additional 45 min in 2% PFA diluted in PBS. Further dissection was done to release the retinas from the RPE, and four nicks were made so that the retina would lay flat. Whole retinas were blocked in 500 mL of PBS containing 0.3% Triton X100% and 3% goat serum for 2 hr at RT. Either mouse anti-Brn3a (Millipore, cat #AB5945, RRID:AB_92154) (1:250), rabbit anti-γ13 (generously provided by Robert Margolskee, RRID: AB_2314434)(1:500), rabbit anti-calretinin (Swant, cat #CR 7699/3 hr, RRID: AB_10000321) (1:500), goat anti-ChAT (Millipore, cat #Ab144p, RRID: AB_262156) (1:200), or mouse anti-SMI-32 (Covance, cat #SMI-32R, RRID: AB_2315331) (1:500) was diluted in blocking solution and incubated overnight at 4°C. Retinas were washed 10 min in three changes of PBS, then placed in 1:500 Alexa Fluor secondary antibody (Invitrogen) overnight at 4°C. Retinas were washed as above and mounted flat on slides in VectaShield (Vector Labs, RRID: AB_2336789). To measure cell density was measured by counting cells in four representative areas of each retina, and density was calculated as the number of cells per $mm^2$.

For SCN cell counts, mice were perfused with cold 4% PFA, brains were dissected out and then cryoprotected in 30% sucrose, frozen in OCT, and 25 μm serial coronal sections containing the SCN were taken. Slides were blocked in PBS containing 0.3% Triton X100 and 3% goat serum for 3 hr at RT and then incubated overnight at 4°C in 1:200 monoclonal mouse immunoglobulin G (IgG) 2b anti-human-HuC/D (Thermofisher, cat# A-21271, RRID: AB_2096358). Slides were then washed in PBS and then incubated in 1:500 goat anti-mouse IgG2b (isotype of secondary antibody is critical (Thermofisher, RRID: AB_429670)) overnight at 4°C. Slides were then washed in PBS and mounted in VectaShield (Vector Labs, RRID: AB_2336789) with DAPI. For counting of DAPI-labeled nuclei and Hu-labeled cell bodies, we used a program that was coded in Mathmatica (Wolfram) and was previously described in *Martinelli et al. (2016)*.

## PNA staining

Mice were deeply anesthetized with 30 ml/kg Avertin followed by cervical dislocation. Eyes were isolated and fixed in 4% PFA for 1 hr. Retinas were dissected out and then incubated for 2 hr in Alexa 488-conjugated lectin peanut agglutinin (PNA) (Invitrogen cat# L21409, RRID: AB_2315178) diluted

1:200 in PBS with 0.3% Triton X100% and 3% goat serum. Retinas were then washed in PBS and mounted in VectaShield (Vector Labs, RRID: AB_2336789).

## In-situ hybridization

Mice were anesthetized with a ketamine/xylazine mixture before transcardially perfusing with 0.9% saline followed by 4% PFA in PBS, pH 7.4. All tissues were post fixed in 4% PFA overnight at 4°C, cryoprotected in 20% sucrose in PBS, frozen in OCT. Compound Embedding Medium (Tissue-Tek), and stored at −75°C. Serial sections (20 μm) were cut on a Hacker cryostat and thaw mounted on Superfrost Plus slides (Thermo Fisher Scientific). Five adjacent sets of sections were prepared from each postnatal age and stored at –20°C. Probes were generated as described in *VanDunk et al. (2011)*. Slides were immersed in 4% PFA, permeabilized with proteinase K, returned to 4% PFA before being washed in 0.1 M triethanolamine-HCl with 0.25% acetic anhydride. Once blocked in hybridization buffer at 65°C slides were incubated in hybridization buffer containing 1–2 μg/ml DIG-labeled anti-sense cRNA overnight at 65°C. Slides were then washed in 2XSSC buffer at 62°C, washed in 0.2XSSC at 65°C, blocked with 10% normal horse serum (NHS) in 0.1M PBS, and incubated in alkaline phosphatase labeled anti-DIG antibody (1:2000 in 10% NHS; Roche, RRID: AB_514497) overnight. Sections were washed and color was visualized using Nitro blue tetrazolium and 5-Bromo-4-chloro-3-indolyl phosphate (Roche). Staining was stopped after visual inspection. Sections were washed, fixed in 4% PFA, and coverslipped in 90% glycerol, Vectashield Mounting Medium (Vector Laboratories, RRID: AB_2336789), or UltraCruz Mounting Media with DAPI (Santa Cruz Biotechnology).

Images were acquired using a Nikon Eclipse 90i microscope, Photometrics Coolsnap HQ2 camera with a Prior Scientific ProScan II motorized translation stage, and acquired in Volocity (PerkinElmer Life and Analytical Sciences). Images were exported as 8bit JPEG or TIFF files. All images were adjusted for clarity by filtering and/or modifying levels, as necessary, in Photoshop (Adobe Systems).

## Enucleations

In order to remove the eyes, P0 mice were placed on ice for 2 min, and then a 1–2 mm incision was made across each eyelid using a sterile scalpel blade. The scalpel blade was then used to puncture the eyes and forceps were used to pull the eyes free of the orbitals. P60 mice were first anesthetized with intraperitoneal injection of 20 mL/kg of Avertin. Fingers were placed on either side of the eye causing it to bulge, a curved pair of scissors was placed between the eye and the skin, and the optic nerve was cut. Bleeding was controlled by orbital pressure. The animal was monitored over the next several days for signs of infection.

## Wheel running behavior

Mice were placed in cages with a 4.5-inch running wheel, and their activity was monitored with Vital-View software (Mini Mitter), and cages were changed at least every 2 weeks. All free-running periods and phase shifts were calculated with ClockLab (Actimetrics).

6-month-old wild-type (n = 18 mice), $Opn4^{LacZ/LacZ}$ (n = 17), $Opn4^{DTA/+}$ (n = 8), $Opn4^{DTA/LacZ}$ (n = 7), $Opn4^{DTA/DTA}$ (n = 7), and $Opn4^{aDTA/aDTA}$ (n = 18) mice were placed in 12:12 LD for 10 days followed by constant darkness for 14 days. For phase-shifting experiments, a subset of wild-type, $Opn4^{LacZ/LacZ}$, $Opn4^{aDTA/aDTA}$ were used. Wild-type (n = 7), $Opn4^{LacZ/LacZ}$ (n = 8), $Opn4^{DTA/+}$ (n = 8), $Opn4^{DTA/LacZ}$ (n = 7), $Opn4^{DTA/DTA}$ (n = 7), and $Opn4^{aDTA/aDTA}$ (n = 8) mice were exposed to a light pulse (500 lx; CT16) for 15 min, after being in constant dark for 14 days.

P0 and P60 ennucleated animals were placed in 12:12 LD for 24 days followed by constant darkness for 14 days.

Dark-reared animals and control mice (raised in 12:12 LD) were placed in constant darkness without any exposure to light. Wheel running behavior was recorded in constant darkness for 1–3 months. Mice were then either given a 3-hr light pulse (LD reared: n = 17; DD reared: n = 12) or placed in 12:12 LD for 14 days and then presented with a 6-hr shift and allowed to re-entrain for 14 days (LD reared: n = 13; DD reared: n = 16). Mice were then returned to constant darkness for 1 month.

## Cholera toxin injections

WT (n = 12), $Opn4^{LacZ/LacZ}$ (n = 6), $Opn4^{aDTA/aDTA}$ (n = 6), $Opn4^{DTA/+}$ (n = 8), $Opn4^{DTA/DTA}$ (n = 10), and $Opn4^{Cre/+}$ (n = 8), and $Opn4^{Cre/+}$; $Pou4f2^{Z-DTA/+}$ (n = 6) mice used for examination of adult central projections were raised in a standard 12:12 LD cycle. 3-6-month-old mice were anaesthetized with 20 mL/kg of Avertin. Eyes were injected intravitreally using a glass pipet with approximately 2 µl of cholera toxin B subunit conjugated with Alexa Fluor 488 (Thermofisher, cat# C34775) or Alexa Fluor 594 (Thermofisher, cat# C22842) using a Harvard Apparatus HL-190 picospritzer. CTB-488 was used at a concentration of 6.25 µg/µL and CTB-594 was used at 5 µg/µL for all injections. Three days after injection, mice were perfused with 4% PFA, and brains were isolated, cryoprotected in 30% sucrose, frozen in OCT, and 40 µm sections were taken using a cryostat. Sections were dried overnight, mounted in VectaShield, and imaged on a Zeiss Imager M1 upright epifluorescence microscope (AxioVision). Retinas were also dissected, mounted in vectashield, and examined for good injection quality.

Mice used to examine LGN projections at P8 were born and raised in 12:12-LD cycle. At P7, mice were anaesthetized on ice, and ocular injections of contrasting fluorescent anterograde tracers (cholera toxin subunit B (CTB)-Alexa-488 and CTB-Alexa-594) were made. At P8, mice were perfused with 4% PFA, and brains were isolated, cryoprotected in 30% sucrose, frozen in OCT, and 40 µm sections were taken using a cryostat. Sections were dried overnight, mounted in VectaShield (Vector Laboratories, RRID: AB_2336789), and imaged on a Zeiss Imager M1 upright epifluorescence microscope (AxioVision). Retinas were also dissected, mounted in vectashield, and examined for good injection quality. Only animals with complete retinal labeling were assessed further. For assessment of SCN innervation at P7, mice were injected with CTB-Alexa-488 at P4. At P7, mice were perfused with 4% PFA, and brains were isolated, cryoprotected in 30% sucrose, frozen in OCT, and 40 µm sections were taken using a cryostat. Sections were dried overnight, mounted in VectaShield (Vector Laboratories, RRID: AB_2336789), and imaged on Zeiss LSM 700 Confocal.

## Quantification of eye-specific segregation

Quantifications were performed with the analyzer blinded to the genotypes being measured.

### Percent overlap between contralateral and ipsilateral projections

40 µm serial sections were taken spanning the dLGN. Images were taken of every section containing the dLGN. In ImageJ, for each image, a background threshold was determined by measuring the maximum intensity of pixels in regions of the section that contained no retinal innervation. Each image was binerized at this intensity threshold. Pixels from contralateral fibers were psedo-colored red and pixels from ipsilateral fibers were psedo-colored green. The whole dLGN was selected. Percent overlap represents ((number of yellow pixels/total number of pixels)*100)), percent ipsilateral represents ((number of green pixels/total number of pixels)*100)), and percent contralateral represents ((number of red pixels/total number of pixels)*100))(*Datwani et al., 2009*; *Demas et al., 2006*). Averages presented in bar graphs represent all sections from the dLGN pooled, and the distribution of segregation throughout the dLGN is shown in *Figure 4—figure supplement 4F*. Representative images were always taken from the region of the dLGN indicated by the blue arrow in *Figure 4—figure supplement 4F*.

### Quantification of eye-specific segregation using the variance of R-value distribution (additional quantification method for eye specific segregation)

A complete series of 40 µm sections spanning the dLGN were digitally imaged with settings (gain, exposure time and gamma level) that were identical for all photomicrographs and for both tracers for all mice. Sections from each brain were aligned to ensure all images analyzed were within the same region of the dLGN. We were blind to genotype until data were processed and analyzed, as described by (*Renna et al., 2011*). For this quantification method, 15 sections from the middle of the LGN were analyzed per adult mouse, per dLGN hemisphere. R-values were calculated for all pixels within each dLGN. This was done by selecting a region of interest that was maximally innervated by the contralateral eye and a region of interest that was maximally innervated by the ipsilateral eye. The R-distribution for maximally ipsilateral segregated pixels and maximally contralateral pixels were calculated and averaged across all 15 dLGN sections. Pixels with an R-value less than 99.9% of the

maximally contra segregated pixels and more than 99.9% of all maximally ipsilateral segregated pixels were termed unsegregated pixels.

Similar to (*Renna et al., 2011*), we calculated the variance for each dLGN R-distribution. Larger variances are indicative of a wider range of values (more ipsilateral dominant pixels and contralateral dominate pixels), and thus, fewer pixels with more balanced left and right eye input.

## Reconstruction of retinal innervation of the SC

Similar to (*Fernandez et al., 2012*), Coronal sections were used for the SC reconstruction using Matlab (Math Work). For each section, the retino-recipient SC was outlined, and the total retinotopic area was calculated. Images were converted to 8-bits of grey scale and the optic density of CTB-staining was calculated. The total length was measured and divided in bins (4 μm) from the medial to lateral region. The CTB density was obtained by dividing the total pixel area by CTB+ pixels. Finally, a colorimetric thermal representation was applied (from 0% = blue to 100% = red). All sections containing the SC were used for a final reconstruction of the retinal projection to the SC.

## Retinal wave recordings

Mice were shipped overnight at postnatal day 5 ($Opn4^{DTA/DTA}$ from Johns Hopkins University and wild-type control mice from Jackson Laboratories). All procedures involving the use of animals were in accordance with the National Institutes of Health and approved by the Brown University Institutional Animal Care and Use Committee. Upon arrival at Brown University, the P6 mice were sacrificed via a lethal intraperitoneal injection of Beuthanasia. As previously described, retinas were extracted and placed ganglion-cell side down on the array under dim red light (*Renna et al., 2011*). The retinas were continuously superfused with oxygenated Ames solution maintained at 36–37°C, and were kept in constant darkness. Recordings were made after at least 30 min of dark adaptation.

The raw analog data were digitized using MC Rack software (Multi Channel Systems) before being filtered. Further processing was done in OfflineSorter (Plexon Inc.). In order to determine the number of neurons being sampled by each electrode, we filtered the raw digitized data using a 125 Hz high-pass filter. A threshold of 5 standard deviations from the baseline voltage was used as the criterion for a spike. The candidate spikes were then sorted using OfflineSorter (Plexon Inc.) for the first two principal components of the waveforms, using a standard T-distribution E-M algorithm (*Shoham et al., 2003*). Spikes appearing within a 1 ms window on 70% of channels were assumed to be artifacts (caused by bubbles or other disturbances) and were discarded.

### Spike time tiling coefficients

To quantify the amount of correlation between two spike trains, we used the spike time tiling coefficient (STTC) (*Cutts and Eglen, 2014*). We calculated the STTC for every pair of units in a retina, using a correlation time interval of $\Delta t = 50$ ms, and took the median over the interelectrode distance to get a STTC vs Interelectrode Distance curve. We also identified units from the ground channel, and units from electrodes that had poor contact with the retina (had a mean firing rate lower than 0.25 Hz over the entire 10 min recording) for later removal.

### Burst identification

Spike timestamps were exported to NeuroExplorer v4.133 (Nex Technologies) for burst analysis. We employed NeuroExplorer's Surprise algorithm to detect bursts of spiking, using a minimum surprise value of 4. This value was based on (*Kirkby et al., 2013*), where spike trains were considered bursting if a Poisson probability dropped below $10^{-4}$, and surprise-values are the negative logarithm of that probability.

### Calculation of firing properties

All spike timestamps and burst start- and end-timestamps from all sorted units were exported to text from NeuroExplorer. Firing properties were calculated from these data files using custom Visual Basic for Applications macros. Our calculations proceeded as follows, for each unit. First, units from invalid channels (as defined above) were excluded from further analysis. Second, bursts shorter than 0.75 s or longer than 15 s were also excluded, although their constituent spikes were kept for later firing rate calculations. Third, bursts that started less than half of the maximum burst duration (7.5 s)

after the recording started or that ended less than that duration before the recording ended were excluded from further analysis. This ensured that interrupted WABs—WABs that occurred while the recording was being started or stopped—were not analyzed. The constituent spike timestamps of these bursts were also kept for later firing rate calculations.

WABs were identified as follows. All bursts were divided into 10 bins of equal duration. For a given burst, if the start of at least one of its bins was within a time interval $\Delta t$ of the start of any bin of a different burst, then those two bursts were said to be 'associated'. The interval $\Delta t$ was equal to half of the sum of the two bursts' bin-durations. If a burst was associated with a burst on at least two neighboring channels, then it was classified as a WAB, where a channel's 'neighboring channels' were any of the eight channels physically surrounding it on the MEA grid. Bursts could not be associated with bursts on the same unit or the same channel, and a burst could not be associated with a burst from multiple units on a single neighboring channel. Finally, spiking property values for an entire retina were taken as the average over all of its units' values.

## Statistics

For spiking properties, we employed two-tailed Student's T-tests assuming unequal variance for each property, with a significance level of $\alpha = 0.05$. We used the Holm-Bonferroni method to control the familywise error rate of these multiple comparisons. To compare the dependency of STTC on distance between these populations, we employed a second round of Student's T-tests, comparing the STTC-values at all interelectrode distances less than or equal to 800 μm.

## Virtual optomotor system

Using the virtual optokinetic system (*Douglas et al., 2005*), we placed individual mice in a box created using four computer screens, which display sine wave gratings that move to create virtual cylinder. If mice can see the gratings, they track the moving bars by turning their head. To obtain an estimate of visual acuity using this setup, we increased the spatial frequency (SF) of the gratings until the mice no longer tracked the movement of the gratings.

## The visual water task

In the visual water task (*Prusky et al., 2000*), mice are place in a trapezoidal shaped maze that contains water and has a divider down the middle to create two arms. At the end of one arm, a gray panel in displayed and in the other arm, sine wave grating is displayed. Mice were trained, using gratings with a low SF, to associate the gratings with a hidden platform that allows them to escape from the water. The location of the grey panel and gratings with the hidden platform were moved between the arms in a pseudorandom pattern that mice cannot memorize. Any entrances into the arm containing the grey panel were recorded as incorrect, after the mice could reliably swim to the platform with greater that 90% accuracy, the SF of the gratings was increased. An animal's threshold was considered to have been reached once they failed to have better than 70% accuracy out of 10 trials. This threshold was confirmed by retesting the previous SF, which is 0.02cycles/degree lower, and then retesting the spatial frequency that the mice initially failed. If the mouse's behavior repeated where they could see the lower SF, but not one increment higher, then the visual acuity of that mouse was recorded as the last spatial frequency where they had better than 70% accuracy.

## Acknowledgements

We would like to thank Marnie Halpern, Carla Shatz, Rejji Kuruvilla, Tiffany Schmidt, and Alan Rupp for their critical reading and advice on the manuscript. We would also like to thank the mouse tri-lab for suggestions and advice. Funding was provided by The Johns Hopkins University-Dean's office funds, The David and Lucile Packard Foundation Fellowship, The Alfred P. Sloan Fellowship, National Institutes of Health Grants R01-GM076430 and R01-EY019053 (to SH); R01-EY017137 (to DMB); DC007395 (to HZ); R01-GM104991 (to EH); R01-HL089742 (to PAG); F32-EY20108 and R15EY026255 (to JMR); and the Canadian Institutes of Health Research MOP-77570 (to MC). In addition, this research was supported (in part) by the Intramural Research Program of the NIMH.

## Additional information

### Funding

| Funder | Grant reference number | Author |
|---|---|---|
| National Institute of General Medical Sciences | GM076430 | Samer Hattar |
| National Eye Institute | R01-EY019053 | Samer Hattar |
| David and Lucile Packard Foundation | | Samer Hattar |
| Alfred P. Sloan Foundation | | Samer Hattar |
| Johns Hopkins University | | Samer Hattar |
| National Eye Institute | R01-EY017137 | David M Berson |
| National Institute on Deafness and Other Communication Disorders | DC007395 | Haiqing Zhao |
| National Institute of General Medical Sciences | R01-GM104991 | Erik D Herzog |
| National Heart, Lung, and Blood Institute | R01-HL089742 | Paul A Gray |
| National Eye Institute | F32-EY20108 | Jordan M Renna |
| National Eye Institute | R15EY026255 | Jordan M Renna |
| Canadian Institutes of Health Research | MOP-77570 | Michel Cayouette |

The funders had no role in study design, data collection and interpretation, or the decision to submit the work for publication.

### Author contributions

KSC, Conceptualization, Data curation, Formal analysis, Supervision, Investigation, Methodology, Writing—original draft, Writing—review and editing; JMR, Conceptualization, Data curation, Formal analysis, Investigation, Methodology, Writing—original draft, Writing—review and editing; DSM, Conceptualization, Data curation, Formal analysis, Investigation, Methodology; DCF, Conceptualization, Resources, Formal analysis, Investigation, Visualization, Diego was the one who did crucial experiments now in Figure 1, to answer the reviewers' concerns about the strength of the DTA versus the aDTA in killing ipRGCs during development; WTK, Conceptualization, Formal analysis, Investigation; MBT, Conceptualization, Investigation; JLE, Conceptualization, Investigation, Methodology; GSL, Data curation, Formal analysis, Investigation; CV, Resources, Data curation, Investigation, Visualization; DCV, Formal analysis, Investigation, Methodology; AT, Resources, Formal analysis, Investigation; SW, Data curation, Investigation; PAG, Conceptualization, Resources, Data curation, Formal analysis, Supervision, Investigation; MC, Resources, Supervision, Funding acquisition, Investigation; EDH, Formal analysis, Supervision, Investigation, Writing—original draft; HZ, Conceptualization, Resources, Formal analysis, Supervision, Investigation, Methodology, Writing—original draft; DMB, Supervision, Funding acquisition, Writing—original draft, Writing—review and editing; SH, Conceptualization, Supervision, Funding acquisition, Writing—original draft, Writing—review and editing

### Author ORCIDs

Kylie S Chew, http://orcid.org/0000-0003-4752-009X
William T Keenan, http://orcid.org/0000-0003-3381-744X
Samer Hattar, http://orcid.org/0000-0002-3124-9525

### Ethics

Animal experimentation: Animals were housed and treated in accordance with NIH and IACUC guidelines, and used protocols approved by the Johns Hopkins University and Brown University Animal Care and Use Committees (Protocol numbers MO16A212 and 1010040).

## Additional files

**Supplementary files**

• Supplementary file 1. Description of all mouse lines.

• Supplementary file 2. Values and statistics for retinal wave recordings. Properties of spontaneous retinal activity in P6 WT and $Opn4^{DTA/DTA}$ mice. Recordings were done in darkness. * represent statistically significant differences after Student's T-tests with a Holm-Bonferroni correction, α = 0.05, m = 22. Significance is determined when the p-value is less than or equal to the Holm-Bonferroni corrected p-value.

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
