## [Decision Letter]

Thank you for submitting your article "200 ipRGCs can regulate both maturation of the circadian clock and segregation of retinogeniculate projections in mice" for consideration by *eLife*. Your article has been reviewed by two peer reviewers, and the evaluation has been overseen by a Reviewing Editor and a Senior Editor. The reviewers have opted to remain anonymous.

The reviewers have discussed the reviews with one another and the Reviewing Editor has drafted this decision to help you prepare a revised submission.

Summary:

This manuscript presents evidence that a small number of intrinsically photosensitive retinal ganglion cells (ipRGCs) regulate development of the suprachiasmatic nucleus (SCN) and eye specific segregation of projections to the geniculate in early development with consequences for the period of circadian clock and visual acuity. The primary evidence for this conclusion comes from comparisons between a number of transgenic mice using different versions of the diphtheria toxin cytotoxic lesion of ipRGCs. They present a new model – opn4DTA – and associated evidence that it has a more complete and earlier onset deletion of ipRGCs than is the case for opn4aDTA animals. In other respects, the phenotype of these two animals as hemizygotes is broadly similar but when homozygous for the toxin, DTA mice have longer circadian period and disrupted segregation of LGN projections compared to aDTA mice. The authors attribute these findings to an early loss of all ipRGCs in the DTA mice. Another model used (*Opn4^Cre^ Brn3b^zDTA^*) allows 200 M1 ipRGCs targeting the SCN to survive. This animal has normal period and a normal thalamic projection. The authors conclude that these few ipRGCs can control development. They show that dark rearing (and early enucleation) also lengthens circadian period but does not phenocopy effects of ipRGC loss on the thalamic projection.

The differences in phenotype between the mouse lines appear robust and the authors are to be complimented on the care taken in their documentation and description. The demonstration that circadian period is impacted by dark rearing and that this effect is so readily reversed is very interesting in itself.

Essential revisions:

1) Enthusiasm for this work is constrained by the limited description of the phenotype of the DTA/DTA mouse. The manuscript is built around a comparison of this mouse with other transgenic lines yet we do not know how complete the lesion of ipRGCs is in this genotype, the developmental time course of their loss, or how specific the lesion is to ipRGCs. Without answers to these questions it is hard to draw definitive conclusions. The authors attribute differences reported between DTA/DTA and aDTA/aDTA or DTA mice to a more complete loss of ipRGCs. This may be a reasonable supposition but we do not know for sure either that there is a more complete loss early in development or that the DTA/DTA does not have off target effects on other important cell types. As the only marker of ipRGCs is melanopsin, there is no way of identifying ipRGCs in the DTA/DTA mice. The only way of measuring ipRGC loss in this genotype is by looking at projections to the SCN. These are lost at 6 months showing that lesion of this projection in adults is more complete, but we do not know whether that is true for other ipRGC types (which express less melanopsin and might therefore be more resistant) or whether the lesion is more complete also at early stages of development. More importantly, this method is also not appropriate for looking at off target effects. It is common for transgenes to have some 'leaky' expression in cells other than those targeted. How confident can the authors be that DTA is not also killing other cells? If it were then this could provide an alternative explanation for the observations.

2) Another important question is whether the DTA mice have the same strain background as other lines? This is not clear from the methods.

3) There are a couple of other instances of over interpretation. The evidence that 200 ipRGCs are sufficient to support development comes from lack of phenotype in *Opn4^Cre^ Brn3b^zDTA^*. To interpret that observation, we need to know the time course of ipRGC loss in this genotype and in DTA/DTA. If brn3b+ve ipRGCs survive longer in *Opn4^Cre^ Brn3b^zDTA^* than in DTA/DTA that would provide an alternative explanation for this effect.

4) The authors show deficits in acuity by optomotor and swim maze tests in DTA/DTA mice compared to wild type and *Opn4^LacZ/LacZ^*mice. They interpret this as an impact of poor eye segregation. That is not the only possible explanation for those data. Loss of ipRGCs could impact acuity because lack of entrainment means that animals were tested at different circadian phases, the pupil is more dilated, direct influences of ipRGCs on retinal function are lost, or some as yet undefined impact of ipRGCs on vision is absent.

5) The authors encourage the reader to pull together the circadian and visual effects as instances of convergence between image-forming (IF) and non-image-forming (NIF) visual systems in development. Are effects on the clock and thalamic projections in any mechanistic sense linked? They both occur in DTA/DTA mice, but while the projection phenotype is clearly an error in optic nerve development, that seems not to be the case for the SCN. The SCN, but not LGN, effect can be phenocopied by dark rearing (unlike the projection deficit) and disappears immediately upon light exposure. As it is rapidly reversible, this not an error in development of anatomy or connectivity (authors present additional evidence that these are normal). If these are two quite separate effects on development how does presenting these in the IF vs NIF dichotomy help the reader to interpret?

6) The Introduction and Abstract present the logic for the study in terms of the growing evidence of blurred distinction between NIF and IF vision in adults and the need to ask whether this also the case in development? This leads the reader to link effects on the clock and thalamus in ways that may not be appropriate (see above). A much more direct motivation for the study is the growing reports that melanopsin and/or ipRGCs regulate retinal development (see papers by Copenhagen/Lang and Berson groups). Building from that literature would better place this work in context of the current state of knowledge.

7) It seems from the data presented that overlap of projections from 2 eyes is reduced in *Opn4^Cre^ Brn3b^zDTA^* compared to wild types. Is this true?

8) The requirement for very few (200) ipRGCs to assist eye-specific segregation may not be very surprising given the dendritic coverage of these cells (big) and the fact that they are a "net", for photoreception, not a high acuity system for detailed analysis. The original "splash" regarding ipRGCs impacting eye-specific segregation (and retinal waves) was published by Renna et al., 2011. The results presented here are different but the novelty resides in the "200" aspect which, while interesting, is hard to set in context. What if it had been 300 or 500? Is the number past 1 relevant?

9) Given the retinal wave phenotype in the Renna et al., 2011 paper, do the authors have any evidence relevant to this phenotype?

[Editors' note: further revisions were requested prior to acceptance, as described below.]

Thank you for resubmitting your work entitled "A subset of ipRGCs regulates both maturation of the circadian clock and segregation of retinogeniculate projections in mice" for further consideration at *eLife*. Your revised article has been favorably evaluated by a Senior editor, a Reviewing editor, and two reviewers.

The manuscript has been improved but there are some remaining issues that need to be addressed before acceptance, as outlined below:

This revision is much improved by additional data dealing with the major issues with the original submission and some changes in presentation. Phenotypes for the various mouse lines are clear and well described. We reiterate however that a weakness of the study design is that two important bits of information for interpreting the data are fundamentally unknowable – what happens to ipRGCs of all types in the DTA/DTA and aDTA/aDTA animals, and whether there are off target effects on other cells in the retina and/or brain of these transgenes. The authors have addressed these questions with more data that are consistent with their preferred interpretation, that the DTA/DTA ablation of ipRGCs occurs earlier and is more extensive than in aDTA/aDTA, and that there is no off target effect. We would prefer a more balanced coverage of these critical questions, the authors present the positive evidence for their interpretation but do not give the limitations of their data appropriate coverage. A paragraph in the Discussion describing what they cannot know and what alternative explanations for their data this allows would be helpful for the reader.

One note to authors: the reply to other reviews has in several places, a "we told you before, in the previous version" tone. This is a source of opaqueness in the paper too; we suggest the authors consider the possibility that the reviewers misunderstood in part because the presentation was not clear and/or there are a lot of conditional mutants. This is not a simple and straightforward study, which requires like any challenging conversation, more clarity, not less. We recommend they consider how the reader will interpret their findings and think hard about striving for more clarity about novelty and methodology for the non-expert.

---

## [Author Response]

*Essential revisions:*

*1) Enthusiasm for this work is constrained by the limited description of the phenotype of the DTA/DTA mouse.*

This is a surprising comment since Figure 1—figure supplement 1 of the original manuscript included a thorough description of the *Opn4^DTA/DTA^* mouse. Nevertheless, as described below, we have now added additional data for the *Opn4^DTA/DTA^* mice.

The manuscript is built around a comparison of this mouse with other transgenic lines yet we do not know how complete the lesion of ipRGCs is in this genotype, the developmental time course of their loss, or how specific the lesion is to ipRGCs. Without answers to these questions it is hard to draw definitive conclusions. The authors attribute differences reported between DTA/DTA and aDTA/aDTA or DTA mice to a more complete loss of ipRGCs. This may be a reasonable supposition but we do not know for sure either that there is a more complete loss early in development or that the DTA/DTA does not have off target effects on other important cell types.

Again, this is a surprising comment. It is true that we have not specifically tested the *Opn4^DTA/DTA^* mice, but we have tested the heterozygous mice extensively and across developmental timecourse (Figure 1). In fact, in the original submission we included data documenting ipRGC loss in mice that expressed, from the melanopsin locus, one copy of the toxin (either aDTA or DTA) and one copy of a marker to identify ipRGCs and their projections (either Cre combined with an alkaline phosphatase reporter, or tau-LacZ). We used mice heterozygous for either aDTA or DTA to establish the extent and timing of ipRGC ablation, which in mice homozygous for the toxin should be either the same or faster and/or more extensive than in mice heterozygous for toxin. However, now, we have added data that directly assess the effect on the homozygous animals. As suggested, we injected CTB in P4 pups, harvested brain tissue at P7, and imaged retinal innervation of the SCN in WT, *Opn4^aDTA/aDTA^*, and *Opn4^DTA/DTA^* mice. These new data are shown in Figure 1 (new panel B), and clearly show the earlier, more substantial ablation of SCN projecting ipRGCs in *Opn4^DTA/DTA^* mice compared to both WT and *Opn4^aDTA/aDTA^*.

*As the only marker of ipRGCs is melanopsin, there is no way of identifying ipRGCs in the DTA/DTA mice. The only way of measuring ipRGC loss in this genotype is by looking at projections to the SCN. These are lost at 6 months showing that lesion of this projection in adults is more complete, but we do not know whether that is true for other ipRGC types (which express less melanopsin and might therefore be more resistant) or whether the lesion is more complete also at early stages of development.*

As described above, to specifically answer the early stages of development loss, we have now carried out the experiment examining SCN innervation in postnatal Opn4^DTA/DTA^ mice (Figure 1 new panel B).

In terms of examining loss of specific ipRGCs subtypes, it is important to note that the reviewers are correct that assessment of ipRGC projections to the SCN specifically provides information regarding ablation of M1 ipRGCs. Determining the timeline of ablation of specific ipRGC subtypes during development is not possible since the subtypes cannot be accurately identified until dendrite arborization is complete, which is between P10-P15 (Coombs, Van Der List and Chalupa, 2007) and after cell loss is complete in Opn4^DTA^ mice (Figure 1). We can make inferences about the proportion M1 and non-M1 ipRGC subtypes that persist based on adult cell counts that were performed in mice heterozygous for toxin and included in the original submission. In mice with one copy of a toxin and one copy of Cre combined with an alkaline phosphatase reporter, we counted the total number of ipRGCs that survive toxin expression (~1500 cells in *Opn4^aDTA/aDTA^* mice and ~500 cells *Opn4^DTA/DTA^* mice). We also counted the number of M1 cells that survive, which is ~150 in aDTA mice and ~75 in DTA mice. Thus, about ~1350 non-M1 ipRGCs persist in mice with one copy of aDTA, and about ~425 non-M1 ipRGCs persist in mice with one of copy DTA. This is consistent with the knowledge that M1 ipRGCs express the most melanopsin and therefore express the most toxin and undergo the most substantial ablation.

However, the reviewers, of course, are asking about loss of specific non-M1 ipRGC subtypes in *Opn4^DTA/DTA^* animals. Without a copy of melanopsin (or a reporter for melanopsin expression), we have no way of labeling the ipRGCs and their dendrites in order to identify the number of remaining ipRGCs of each subtype. Thus, to address this question, we must use molecular markers for ipRGCs subtypes. While the field has been attempting to discover molecular markers for each ipRGC subtype for many years now, there has been minimal success. However, one marker, which identifies the M4 subtype, has been discovered: a protein called SMI-32. SMI-32 was historically a marker of ON and OFF α ganglion cells. With the realization that ON α cells are one and the same as M4 ipRGCs, a marker for M4 ipRGC was identified (Estevez et al., 2012; Schmidt et al., 2014). It is worth noting that all M4 cells are SMI-32 positive, but since SMI-32 also labels OFF α cells, about 50% of SMI-32 positive ganglion cells are melanopsin expressing (Schmidt et al., 2014). We actually included SMI-32 staining in WT and *Opn4^DTA/DTA^* mice in the original submission, but did not highlight that it was also labeling M4 ipRGCs. We have done so now. We detected no difference in the number of SMI-32 positive cells in DTA/DTA compared to WT controls. Thus, many, if not all, M4 ipRGCs persist in *Opn4^DTA/DTA^* mice, which is consistent with there still being AP positive ipRGCs in *Opn4^Cre/DTA^*; Z/AP mice and M4 ipRGCs expressing less melanopsin than M1s (Sexton and Van Gelder 2015). We have included a new section in the Results to describe this point.

The major contribution of our data is the demonstration that Brn3b negative M1 ipRGCs are important for both the maturation of the circadian clock and eye-specific segregation. We now have demonstrated beyond reasonable doubt that these cells are ablated in *Opn4^DTA/DTA^* mice. Thus, we feel that an exhaustive assessment of which non-M1 ipRGCs die in the *Opn4^aDTA^* and *Opn4^DTA^* lines does not further augment the conclusions of this of study.

*More importantly, this method is also not appropriate for looking at off target effects. It is common for transgenes to have some 'leaky' expression in cells other than those targeted. How confident can the authors be that DTA is not also killing other cells? If it were then this could provide an alternative explanation for the observations.*

Off-target cell loss due to leaky expression of DTA was initially also a major concern for us, and thus, we conducted a myriad of experiments to prove to ourselves, and hopefully future readers, that this concern is unwarranted. We actually included much of this data in the original submission (shown in current Figure 1—figure supplement 1). We have now include additional data (see Figure 1—figure supplement 1 and Figure 1—figure supplement 2). We have also expanded our description of these data in the main text of the manuscript. Since it is obviously not possible to account for every individual cell in the whole organism, we can most generally report that the *Opn4^DTA/DTA^* mice are viable, fertile, and do not exhibit any gross abnormalities in size. This information has been added to the paper. More specifically, we focused on evaluating the retina and the brain regions most relevant to this study. General retinal structure was assessed by staining retinal sections from WT and *Opn4^DTA/DTA^* mice with H&E stain and additionally fluorescent staining for various retinal cells type, including cones, ON bipolar cells, calretinin positive amacrine and ganglion cells, and Brn3a positive ganglion cells. We have now quantified the total retinal thickness as well as the thickness of each of the retinal layers of WT and *Opn4^DTA/DTA^* based on H&E staining (see new Figure 1—figure supplement 1). Comparison of WT and *Opn4^DTA/DTA^* retinas showed no difference in total retinal thickness or the thickness of any of the individual retinal layers. We also counted the number of SMI-32 positive ganglion cells, Brn3a positive ganglion cells, cones, and starburst amacrine cells. When we compared WT and *Opn4^DTA/DTA^* mice, there was no difference in cell number of any of these cell types. Also, as mentioned above, about 50% of SMI-32 positive ganglion cells are actually M4 ipRGCs, which means that expression of two copies of DTA from the melanopsin locus does not even ablate all ipRGCs. This fact helped convince us that there is no off-target killing of non-ipRGC cells. Nonetheless, we did further tests to verify there were no off-target effects outside the retina. We counted the total number of nuclei (based on DAPI staining) and the total number of neurons (identified by antibody staining for Hu) in the SCN of WT and *Opn4^DTA/DTA^* mice. There was no difference between WT and *Opn4^DTA/DTA^* in the number of nuclei or neurons when compared in total or section by section through the SCN. These new data are in new Figure 1—figure supplement 2. We have now also measured the total size of the dLGN, which was also not different between WT and *Opn4^DTA/DTA^* mice, or between any other tested mouse line. These new data are in Figure 4—figure supplement 4 new panel E. We hope the reviewers agree that the combination of the data presented in the original submission and our new measurements thoroughly and rigorously demonstrates there is no off-target killing of non-ipRGCs in *Opn4^DTA/DTA^* mice.

*2) Another important question is whether the DTA mice have the same strain background as other lines? This is not clear from the methods.*

The reviewer is correct that this is an important point. We apologize for not including this information in the original submission. We strived to have all the mice on similar strain backgrounds as possible; although, it is challenging to have identical backgrounds considering the number of crosses performed for this study. All mice were either on C56BL/6J or a C56BL/6J – 129/J hybrid background. This information has been added to mouse strain table in the supplementary information (See updated table of mouse lines). We would like to stress that we always strain matched the control group and the most meaningful comparisons (for example WT, *Opn4^DTA^*, dark reared, and enucleated mice were all on the pure B6 background). In terms of circadian biology the C56BL/6 and 129/J strains are very similar (Schwartz and Zimmerman 1990), and there have been no reported differences in eye specific segregation nor did we observe any differences between pure C56BL/6J and mice on a mixed background.

*3) There are a couple of other instances of over interpretation. The evidence that 200 ipRGCs are sufficient to support development comes from lack of phenotype in Opn4^Cre^ Brn3b^zDTA^. To interpret that observation, we need to know the time course of ipRGC loss in this genotype and in DTA/DTA. If brn3b+ve ipRGCs survive longer in Opn4^Cre^ Brn3bzDTA than in DTA/DTA that would provide an alternative explanation for this effect.*

We actually included the developmental timecourse of ablation in the *Opn4^Cre/+^;Brn3b^zDTA/+^* (now called *Opn4^Cre/+^; Pou4f2^zDTA/+^* in the revised manuscript) animals in the original submission (Figure 4—figure supplement 5) and showed that ipRGC ablation is already complete by P7. Thus, the timing brn3b positive ipRGCs cell loss is comparable in DTA and *Opn4^Cre/+^; Pou4f2^zDTA/+^* mice. ipRGC ablation may even be a bit faster in *Opn4^Cre/+^; Pou4f2^zDTA/+^* mice and is also known to include ablation of the M4 population (Schmidt et al., 2014). In fact, more ipRGCs are ablated in these animals than the DTA line and yet no segregation deficits are observed in the *Opn4^Cre/+^; Pou4f2^zDTA/+^* animals.

*4) The authors show deficits in acuity by optomotor and swim maze tests in DTA/DTA mice compared to wild type and Opn4^LacZ/LacZ^mice. They interpret this as an impact of poor eye segregation. That is not the only possible explanation for those data. Loss of ipRGCs could impact acuity because lack of entrainment means that animals were tested at different circadian phases, the pupil is more dilated, direct influences of ipRGCs on retinal function are lost, or some as yet undefined impact of ipRGCs on vision is absent.*

The reviewers make a good point here, and thus, we have added a discussion of alternative explanations regarding the reduction in acuity. However, the loss of acuity in *Opn4^DTA/DTA^* mice cannot be explained by a loss of photoentrainment because *Opn4^aDTA/aDTA^* mice, which also free run, do not exhibit such a substantial reduction in acuity. *Opn4^DTA/+^* mice, which do photoentrain also exhibited deficits in visual acuity. Furthermore, when the pupil is fully dilated with Carbocal there is only a minor reduction in acuity and it is not comparable to the substantial deficits that we observed in *Opn4^DTA/DTA^* and *Opn4^DTA/+^* mice. Nonetheless, direct effects of ipRGCs on retinal functions, including the role of M1 ipRGCs in regulating dopamine in the retina, could contribute. We have now discussed these points in the manuscript.

*5) The authors encourage the reader to pull together the circadian and visual effects as instances of convergence between image-forming (IF) and non-image-forming (NIF) visual systems in development. Are effects on the clock and thalamic projections in any mechanistic sense linked? They both occur in DTA/DTA mice, but while the projection phenotype is clearly an error in optic nerve development, that seems not to be the case for the SCN. The SCN, but not LGN, effect can be phenocopied by dark rearing (unlike the projection deficit) and disappears immediately upon light exposure. As it is rapidly reversible, this not an error in development of anatomy or connectivity (authors present additional evidence that these are normal). If these are two quite separate effects on development how does presenting these in the IF vs NIF dichotomy help the reader to interpret?*

We presented the IF vs NIF dichotomy because we were very surprised that the population of ipRGCs (the Brn3b negative M1 ipRGCs), which project exclusively to NIF brain regions, are also sufficient for the refinement of retinal ganglion cell projections the LGN. This single population appears to have both light dependent functions (circadian period setting) and light independent functions (modulation of spontaneous retinal activity; see point 9). We feel this is a critical point from our work; however, we agree that we could make this much clearer and that our presentation in the original submission could mislead readers. We do not want readers to think there is mechanistic overlap between circadian period setting and refinement of eye-specific segregation, but rather that a single cell type acts through very different mechanisms (light dependent and light independent) to affect the development of both the NIF and IF systems. We think that the addition of retinal wave recordings (see point 9) will help make this clearer, but we have also modified the Introduction and Discussion to hopefully better highlight the conclusions that we aim to make.

*6) The Introduction and Abstract present the logic for the study in terms of the growing evidence of blurred distinction between NIF and IF vision in adults and the need to ask whether this also the case in development? This leads the reader to link effects on the clock and thalamus in ways that may not be appropriate (see above). A much more direct motivation for the study is the growing reports that melanopsin and/or ipRGCs regulate retinal development (see papers by Copenhagen/Lang and Berson groups). Building from that literature would better place this work in context of the current state of knowledge.*

This is an excellent point, and we now have modified the Introduction with the established roles for ipRGCs in retinal development in mind.

*7) It seems from the data presented that overlap of projections from 2 eyes is reduced in opn4cre brn3bzdta compared to wild types. Is this true?*

There is no statistical difference by a one-way ANOVA and this is now stated in the figure legend.

*8) The requirement for very few (200) ipRGCs to assist eye-specific segregation may not be very surprising given the dendritic coverage of these cells (big) and the fact that they are a "net", for photoreception, not a high acuity system for detailed analysis. The original "splash" regarding ipRGCs impacting eye-specific segregation (and retinal waves) was published by Renna et al., 2011. The results presented here are different but the novelty resides in the "200" aspect which, while interesting, is hard to set in context. What if it had been 300 or 500? Is the number past 1 relevant?*

Our point is not that 200 cells is a magic number, but rather that the subset of ipRGCs, which project exclusively to circadian brain regions (the SCN and the IGL) and have intraretinal axonal collaterals, is sufficient to mediate circadian clock maturation and refinement of retinal ganglion cell projections to the LGN. The role of light in the maturation of the circadian clock is surprising; however, the role of this subset of ipRGCs in that process is logical, since they obviously project to the SCN. Yet, the experiments with *Opn4^Cre/+^; Pou4f2^zDTA/+^* mice are critical to show that ipRGCs are not functioning within the LGN to mediate eye-specific segregation (see point 9 below). In the original submission, we implied that this subset must be acting within the retina to mediate eye-specific segregation. With the addition of recordings of retinal waves in WT and *Opn4^DTA/DTA^* mice (see next point) we have corroborated this conclusion. We have also modified the text to better emphasize that 200 cells is not the key point, but rather one specialized subset mediates both of these seemingly distinct development processes.

*9) Given the retinal wave phenotype in the Renna et al., 2011 paper, do the authors have any evidence relevant to this phenotype?*

Renna et al. demonstrated that light (detect by melanopsin) alters duration of bursting during retinal waves. However, independent of light ipRGCs also depolarized during retinal waves. Thus, we have now included an assessment of retinal wave in darkness in P6 WT and DTA/DTA mice using multielectrode recordings. These data are now presented in new Figure 5 and raw numbers and p values are shown in new [Supplementary-material SD2-data]. Spiking properties of RGCs during waves were significantly altered in DTA/DTA mice. Wave-associated bursts were significantly longer in duration, and had higher firing rates, more total spikes and shorter inter-burst intervals than they did in wild-type controls. There were also significantly more spikes outside of bursts. Correlated spiking activity between neurons was very similar between the genotypes, as measured by the spike time tiling. These data indicate that ipRGCs are critical for normal retinal wave activity, even in darkness, and together with our anatomical findings, allow us to suggest that ipRGCs mediate eye-specific segregation of RGC projections to the dLGN by regulating the spiking properties of conventional RGCs during retinal waves.

[Editors' note: further revisions were requested prior to acceptance, as described below.]

*This revision is much improved by additional data dealing with the major issues with the original submission and some changes in presentation. Phenotypes for the various mouse lines are clear and well described. We reiterate however that a weakness of the study design is that two important bits of information for interpreting the data are fundamentally unknowable – what happens to ipRGCs of all types in the DTA/DTA and aDTA/aDTA animals, and whether there are off target effects on other cells in the retina and/or brain of these transgenes. The authors have addressed these questions with more data that are consistent with their preferred interpretation, that the DTA/DTA ablation of ipRGCs occurs earlier and is more extensive than in aDTA/aDTA, and that there is no off target effect. We would prefer a more balanced coverage of these critical questions, the authors present the positive evidence for their interpretation but do not give the limitations of their data appropriate coverage. A paragraph in the Discussion describing what they cannot know and what alternative explanations for their data this allows would be helpful for the reader.*

We have accepted the suggestion and have added the following paragraph to the Discussion:

“A key method in this study was the use of two different versions of diphtheria toxin subunit A that have different potencies, the attenuated versus the full strength forms. [...] Together, although we cannot fully eliminate the possibility of off target effects of the DTA line, we believe this is unlikely.”

*One note to authors: the reply to other reviews has in several places, a "we told you before, in the previous version" tone. This is a source of opaqueness in the paper too; we suggest the authors consider the possibility that the reviewers misunderstood in part because the presentation was not clear and/or there are a lot of conditional mutants. This is not a simple and straightforward study, which requires like any challenging conversation, more clarity, not less. We recommend they consider how the reader will interpret their findings and think hard about striving for more clarity about novelty and methodology for the non-expert.*

We truly apologize if we presented our responses as “we told you before.” We agree that this is a large study with many different mutant lines and manipulations, in addition to covering two different areas, circadian and vision. In our previous response, we aimed to highlight the areas where the concerns raised by the reviewers have been addressed in our original submission. We certainly strive to improve the clarity of our publications. Therefore, we have made several changes throughout the manuscript, which we hope improve the clarity of this study.